# META-LEARNING TRANSFERABLE REPRESENTATIONS WITH A SINGLE TARGET DOMAIN

## ABSTRACT

Recent works found that fine-tuning and joint training—two popular approaches for transfer learning—do not always improve accuracy on downstream tasks. First, we aim to understand more about when and why fine-tuning and joint training can be suboptimal or even harmful for transfer learning. We design semi-synthetic datasets where the source task can be solved by either source-specific features or transferable features. We observe that (1) pre-training may not have incentive to learn transferable features and (2) joint training may simultaneously learn source-specific features and overfit to the target. Second, to improve over fine-tuning and joint training, we propose **M**eta **R**epresentation **L**earni**n**g (MeRLin) to learn transferable features. MeRLin meta-learns representations by ensuring that a head fit on top of the representations with target training data also performs well on target validation data. We also prove that MeRLin recovers the target ground-truth model with a quadratic neural net parameterization and a source distribution that contains both transferable and source-specific features. On the same distribution, pre-training and joint training provably fail to learn transferable features. MeRLin empirically outperforms previous state-of-the-art transfer learning algorithms on various real-world vision and NLP transfer learning benchmarks.

## 1 INTRODUCTION

Transfer learning—transferring knowledge learned from a large-scale source dataset to a small target dataset—is an important paradigm in machine learning (Yosinski et al., 2014) with wide applications in vision (Donahue et al., 2014) and natural language processing (NLP) (Howard & Ruder, 2018; Devlin et al., 2019). Because the source and target tasks are often related, we expect to be able to learn features that are transferable to the target task from the source data. These features may help learn the target task with fewer examples (Long et al., 2015; Tamkin et al., 2020).

Mainstream approaches for transfer learning are fine-tuning and joint training. Fine-tuning initializes from a model pre-trained on a large-scale source task (e.g., ImageNet) and continues training on the target task with a potentially different set of labels (e.g., object recognition (Wang et al., 2017; Yang et al., 2018; Kolesnikov et al., 2019), object detection (Girshick et al., 2014), and segmentation (Long et al., 2015; He et al., 2017)). Another enormously successful example of fine-tuning is in NLP: pre-training transformers and fine-tuning on downstream tasks leads to state-of-the-art results for many NLP tasks (Devlin et al., 2019; Yang et al., 2019). In contrast to the two-stage optimization process of fine-tuning, joint training optimizes a linear combination of the objectives of the source and the target tasks (Kokkinos, 2017; Kendall et al., 2017; Liu et al., 2019b).

Despite the pervasiveness of fine-tuning and joint training, recent works uncover that they are not always panaceas for transfer learning. Geirhos et al. (2019) found that the pre-trained models learn the texture of ImageNet, which is biased and not transferable to target tasks. ImageNet pre-training does not necessarily improve accuracy on COCO (He et al., 2018), fine-grained classification (Kornblith et al., 2019), and medical imaging tasks (Raghu et al., 2019). Wu et al. (2020) observed that large model capacity and discrepancy between the source and target domain eclipse the effect of joint training. Nonetheless, we do not yet have a systematic understanding of what makes the successes of fine-tuning and joint training inconsistent.

The goal of this paper is two-fold: (1) to understand more about when and why fine-tuning and joint training can be suboptimal or even harmful for transfer learning; (2) to design algorithms that overcome the drawbacks of fine-tuning and joint training and consistently outperform them.

To address the first question, we hypothesize that fine-tuning and joint training do not have incentives to prefer learning transferable features over source-specific features, and thus their capability of learning transferable features is rather accidental depending on the property of the datasets. To empirically analyze the hypothesis, we design a semi-synthetic dataset that contains artificially-amplified transferable features and source-specific features simultaneously in the source data. Both the transferable and source-specific features can solve the source task, but only transferable features are useful for the target. We analyze what features fine-tuning and joint training will learn. See Figure 1 for an illustration of the semi-synthetic experiments. We observed following failure patterns of fine-tuning and joint training on the semi-synthetic dataset.

- Pre-training may learn non-transferable features that don't help the target when both transferable and source-specific features can solve the source task, since it's oblivious to the target data. When the dataset contains source-specific features that are more convenient for neural nets to use, pre-training learns them; as a result, fine-tuning starting from the source-specific features does not lead to improvement.

- Joint training learns source-specific features and overfits on the target. A priori, it may appear that the joint training should prefer transferable features because the target data is present in the training loss. However, joint training easily overfits to the target especially when the target dataset is small. When the source-specific features are the most convenient for the source, joint training simultaneously learns the source-specific features and memorizes the target dataset.

Toward overcoming the drawbacks of fine-tuning and joint training, we first note that any proposed algorithm, unlike fine-tuning, should use the source and the target simultaneously to encourage extracting shared structures. Second and more importantly, we recall that good representations should enable generalization: we should not only be able to fit a target head with the representations (as joint training does), but the learned head should also generalize well to a held-out target dataset. With this intuition, we propose **Me**ta **R**epresentation **Learn**ing (**MeRLin**) to encourage learning transferable and generalizable features: we meta-learn a feature extractor such that the head fit to a target training set performs well on a target validation set. In contrast to the standard model-agnostic meta-learning (MAML) (Finn et al., 2017), which aims to learn prediction models that are adaptable to multiple target tasks from multiple source tasks, our method meta-learns transferable representations with only *one source and one target domain*.

Empirically, we first verify that MeRLin learns transferable features on the semi-synthetic dataset. We then show that MeRLin outperforms state-of-the-art transfer learning baselines in real-world vision and NLP tasks such as ImageNet to fine-grained classification and language modeling to GLUE.

Theoretically, we analyze the mechanism of the improvement brought by MeRLin. In a simple two-layer quadratic neural network setting, we prove that MeRLin recovers the target ground truth with only limited target examples whereas both fine-tuning and joint training fail to learn transferable features that can perform well on the target.

In summary, our contributions are as follows. (1) Using a semi-synthetic dataset, we analyze and diagnose when and why fine-tuning and joint training fail to learn transferable representations. (2) We design a meta representation learning algorithm (MeRLin) which outperforms state-of-the-art transfer learning baselines. (3) We rigorously analyze the behavior of fine-tuning, joint training, and MeRLin on a special two-layer neural net setting.

## 2 SETUP AND PRELIMINARIES

In this paper, we study *supervised transfer learning*. Consider an input-label pair $(x, y) \in \mathbb{R}^d \times \mathbb{R}$. We are provided with a source distributions $\mathcal{D}_s$ and a target distribution $\mathcal{D}_t$ over $\mathbb{R}^d \times \mathbb{R}$. The source dataset $\widehat{\mathcal{D}}_s = \{x_i^s, y_i^s\}_{i=1}^{n_s}$ and the target dataset $\widehat{\mathcal{D}}_t = \{x_i^t, y_i^t\}_{i=1}^{n_t}$ consist of $n_s$ i.i.d. samples from $\mathcal{D}_s$ and $n_t$ i.i.d. samples from $\mathcal{D}_t$ respectively. Typically $n_s \gg n_t$. We view a predictor as a composition of a feature extractor $h_\phi : \mathbb{R}^d \to \mathbb{R}^m$ parametrized by $\phi \in \mathbf{\Phi}$, which is often a deep neural net, and a head classifier $g_\theta : \mathbb{R}^m \to \mathbb{R}$ parametrized by $\theta \in \mathbf{\Theta}$, which is often linear. That is, the final prediction is $f_{\theta,\phi}(x) = g_\theta(h_\phi(x))$. Suppose the loss function is $\ell(\cdot, \cdot)$, such as cross entropy loss for classification tasks. Our goal is to learn an accurate model on the target domain $\mathcal{D}_t$.

Since the label sets of the source and target tasks can be different, we usually learn two heads for the source task and the target task separately, denoted by $\theta_s$ and $\theta_t$, with a shared feature extractor

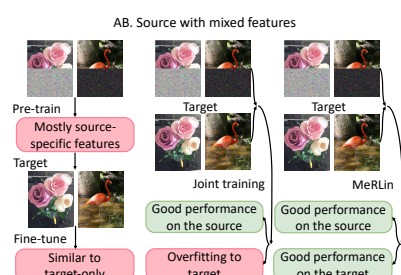
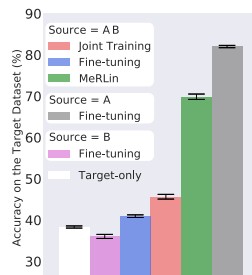

Figure 1: Comparison of fine-tuning, joint training, and MeRLin on the semi-synthetic dataset. **Left**: The semi-synthetic dataset and the qualitative observations on the representations learned by three algorithms. **Right:** Quantitative results on the target test accuracy. See more interpretations, analysis, and results in Section 3.

$\phi$. Let $L_{\widehat{D}}(\theta, \phi)$ be the empirical loss of model $g_\theta(h_\phi(x))$ on the empirical distribution $\widehat{D}$, that is, $L_{\widehat{D}}(\theta, \phi) := \mathbb{E}_{(x,y) \in \widehat{D}} \ell(g_\theta(h_\phi(x)), y)$ where $(x, y) \in \widehat{D}$ means sampling uniformly from the dataset $\widehat{D}$. Using this notation, the standard supervised loss on the source (with the source head $\theta_s$) and loss on the target (with the target head $\theta_t$) can be written as $L_{\widehat{\mathcal{D}}_s}(\theta_s, \phi)$ and $L_{\widehat{\mathcal{D}}_t}(\theta_t, \phi)$ respectively.

We next review mainstream transfer learning baselines and describe them in our notations.

**Target-only** is the trivial algorithm that only trains on the target data $\widehat{\mathcal{D}}_t$ with the objective $L_{\widehat{\mathcal{D}}_t}(\theta_t, \phi)$ starting from random initialization. With insufficient target data, target-only is prone to overfitting.

**Pre-training** starts with random initialization and *pre-trains* on the source dataset with objective function $L_{\widehat{\mathcal{D}}_s}(\theta_s, \phi)$ to obtain the pre-trained feature extractor $\hat{\phi}_{\text{pre}}$ and head $\hat{\theta}_s$.

**Fine-tuning** initializes the target head $\theta_t$ randomly and initializes the feature extractor $\phi$ by $\hat{\phi}_{\text{pre}}$ obtained in pre-training, and *fine-tunes* $\phi$ and $\theta_t$ on the target by optimizing $L_{\widehat{\mathcal{D}}_t}(\theta_t, \phi)$ over both $\theta_t$ and $\phi$. Note that in this paper, fine-tuning refers to fine-tuning all layers by default.

**Joint training** starts with random initialization, and trains on the source and target dataset jointly by optimizing a linear combination of their objectives over the heads $\theta_s$, $\theta_t$ and the shared feature extractor $\phi$: $\min_{\theta_s, \theta_t, \phi} L_{\text{joint}}(\theta_s, \theta_t, \phi) := (1 - \alpha) L_{\widehat{\mathcal{D}}_s}(\theta_s, \phi) + \alpha L_{\widehat{\mathcal{D}}_t}(\theta_t, \phi)$. The hyper-parameter $\alpha$ is used to balance source training and target training. We use cross-validation to select optimal $\alpha$.

## 3 LIMITATIONS OF FINE-TUNING AND JOINT TRAINING: ANALYSIS ON SEMI-SYNTHETIC DATA

Previous works (He et al., 2018; Wu et al., 2020) have observed cases when fine-tuning and joint training fail to improve over target-only. Our hypothesis is that both pre-training and joint training do not have incentives to prefer learning transferable features over source-specific features, and thus the performance of fine-tuning and joint training rely on whether the transferable features happen to be the best features for predicting the source labels. Validating this hypothesis on real datasets is challenging, if not intractable—it's unclear what's the precise definition or characterization of transferable features and source-specific features. Instead, we create a semi-synthetic dataset where transferable features and source-specific features are prominent and well defined.

**A semi-synthetic dataset.** The target training dataset we use is a uniformly-sampled subset of the CIFAR-10 training set of size 500. The target test dataset is the original CIFAR-10 test set. The source dataset of size 49500, denoted by AB, is created as follows. The upper halves of the examples are the upper halves of the CIFAR-10 images (excluding the 500 example used in target). The lower halves contain a signature pattern that strongly correlates with the class label: for class $c$, the pixels of the lower half are drawn i.i.d. from gaussian distribution $\mathcal{N}(c/10, 0.2^2)$. Therefore, averaging the pixels in the lower half of the image can reveal the label because the noise will get averaged out. The benefit of this dataset is that any features related to the top half of the images can be defined as transferable features, whereas the features related to the bottom half are source-specific. Moreover, we can easily tell which features are used by a model by testing the performance on images with masked top or bottom half. For analysis and comparison, we define A to be the dataset that contains the top half of dataset AB and zeros out the bottom half, and B vice versa. See Figure 1 (left) for an illustration of the datasets. Further details are deferred to Section A.1

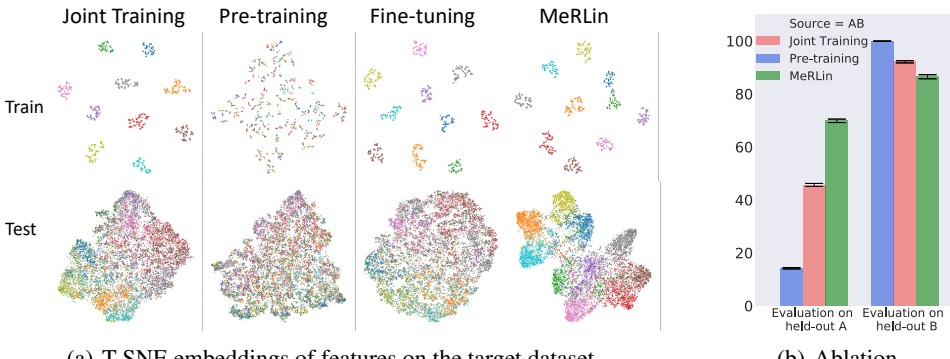

(a) T-SNE embeddings of features on the target dataset.      (b) Ablation.

Figure 2: (a) **T-SNE visualizations of features on the target train and test set**. The representations of pre-training work poorly on both target train and test set, indicating that transferable features are not learned. Both joint training and fine-tuning work well on the target train set but poorly on the test set, indicating overfitting. MeRLin works well on the target test set. (b) **Evaluation of different methods on A and B.** Joint-training and pre-training rely heavily on the source-specific feature B and learn the transferable feature A poorly compared to MeRLin. See more details in Section 3.

In Figure 1 (right), we evaluate various algorithms' performance on target test data. In Figure 2(a) (left), we run algorithms with AB being the source dataset and visualize the learned features on the target training dataset and target test dataset to examine the generalizability of the features. In Figure 2(a) (right), we evaluate the algorithms on the held-out version of dataset A and B to examine what features the algorithms learn. ResNet-32 (He et al., 2016) is used for all settings.

**Analysis:** First of all, *target-only* has low accuracy (38%) because the target training set is small. Except when explicitly mentioned, all the discussions below are about algorithms on the source AB.

*Fine-tuning fails because pre-training does not prefer to learn transferable features and fine-tuning overfits.* Figure 2(b) (pre-training) shows that the pre-trained model has near-trivial accuracy on held-out A but near-perfect accuracy on held-out B, indicating that it solely relies on the source-specific feature (bottom half) and does not learn transferable features. Figure 2(a) (pre-training) shows that indeed pre-trained features do not have even correlation with target training and test sets. Figure 2(a) (fine-tuning) shows that fine-tuning improves the features' correlation with the training target labels but it does not generalize to the target test because of overfitting. The performance of fine-tuning (with source =AB) in Figure 1 (right) also corroborates the lack of generalization.

*Joint training fails because it simultaneously learns mostly source-specific features and features that overfit to the target.* Figure 2(b) (joint training) shows that the joint training model performs much better on held-out B (with 92% accuracy) than on the held-out A (with 46% accuracy), indicating it learns the source-specific feature very well but not the transferable features. The next question is what features joint training relies on to fit the target training labels. Figure 2(a) shows strong correlation between joint training model's features and labels on the target training set, but much less correlation on the target test set, suggesting that the joint training model's feature extractor, applied on the target data (which doesn't have source-specific features), overfits to the target training set. This corroborates the poor accuracy of joint training on the target test set (Figure 1), which is similar to target-only's.[1]

In Section 5, we rigorously analyze the behavior of these algorithms on a more simplified settings and show that the phenomena above can theoretically occur.

## 4    MERLIN: META REPRESENTATION LEARNING

In this section, we design a meta representation learning algorithm that encourages the discovery of transferable features. As shown in the semi-synthetic experiments, fine-tuning does not have any incentive to learn transferable features if they are not the most convenient for predicting the source labels because it is oblivious to target data. Thus we have to use the source and target together to

---

[1]As sanity checks, when the source contains only transferable features (Figure 1, right, source = A), fine-tuning works well, and when no transferable features (Figure 1, right, source = B), it does not.

learn transferable representations. A natural attempt would have been joint training, but it overfits to the target when the target data is scarce as shown in the t-SNE visualizations in Figure 2(a).

To fix the drawbacks of the joint training, we recall that good representations should not only work well for the target *training* set but also *generalize* to the target distribution. More concretely, a good representation $h_\phi$ should enable the generalization of the linear head learned on top of it—a linear head $\theta$ that is learned by fixing the feature $h_\phi(x)$ as the inputs should generalize well to a held-out dataset. We design a bi-level optimization objective to learn such features, inspired by meta-learning for fast adaptation (Finn et al., 2017) and learning-to-learn for automatic hyperparameter optimization (Maclaurin et al., 2015; Thrun & Pratt, 2012) (more discussions below.)

We first split the target training set $\widehat{\mathcal{D}}_t$ randomly into $\widehat{\mathcal{D}}_t^{\mathrm{tr}}$ and $\widehat{\mathcal{D}}_t^{\mathrm{val}}$. Given a feature extractor $\phi$, let $\widehat{\theta}_t(\phi)$ be the linear classifier learned by using $h_\phi(x)$ as the inputs on the dataset $\widehat{\mathcal{D}}_t^{\mathrm{tr}}$.

$$\widehat{\theta}_t(\phi) = \arg\min_\theta L_{\widehat{\mathcal{D}}_t^{\mathrm{tr}}}(\theta, \phi) \tag{1}$$

Note that $\widehat{\theta}_t(\phi)$ depends on the choice by $\phi$ (and is almost uniquely decided by it because the objective is convex in $\theta$.) As alluded before, our final objective involves the generalizability of $\widehat{\theta}_t(\phi)$ to the held-out dataset $\widehat{\mathcal{D}}_t^{\mathrm{val}}$:

$$L_{\mathrm{meta},t}(\phi) = L_{\widehat{\mathcal{D}}_t^{\mathrm{val}}}(\widehat{\theta}_t(\phi), \phi) = \mathbb{E}_{(x,y)\in\widehat{\mathcal{D}}_t^{\mathrm{val}}} \ell(g_{\widehat{\theta}_t(\phi)}(h_\phi(x)), y) \tag{2}$$

The final objective is a linear combination of $L_{\mathrm{meta},t}(\phi)$ with the source loss

$$\underset{\phi\in\boldsymbol{\Phi},\theta_s\in\boldsymbol{\Theta}}{\mathrm{minimize}} \; L_{\mathrm{meta}}(\phi, \theta_s) := L_{\widehat{\mathcal{D}}_s}(\theta_s, \phi) + \rho \cdot L_{\mathrm{meta},t}(\phi) \tag{3}$$

To optimize the objective, we can use standard bi-level optimization technique as in learning-to-learn approaches. We also design a sped-up version of MeRLin by changing the loss to squared loss so that the $\widehat{\theta}_t(\phi)$ has an analytical solution. More details are provided in Section A.3 (Algorithm 2).

*Comparison to other meta-learning work.* The key distinction of our approach from MAML (Finn et al., 2017) and other meta-learning algorithms (e.g., (Nichol et al., 2018a; Bertinetto et al., 2019)) is that we only have a single source task and a single target task. Recent work (Raghu et al., 2020) argues that feature reuse is the dominating factor of the effectiveness of MAML. In our case, the training target task is exactly the same as the test task, and thus the only possible contributing factor is a better-learned representation instead of fast adaptation. Our algorithm is in fact closer to the work on hyperparameter optimization (Maclaurin et al., 2015; Zoph & Le, 2016)—if we view the parameters of the head $\theta_t$ as $m$ **hyperparameters** and view $\phi$ and $\theta_s$ as the ordinary parameters, then our algorithm is tuning hyperparameters on the validation set using gradient descent.

### 4.1 MeRLin Learns Transferable Features on Semi-Synthetic Dataset

We verify that MeRLin learns transferable features in the semi-synthetic setting of Section 3 where fine-tuning and joint training fail. Figure 1 (right) shows that MeRLin outperforms fine-tuning and joint training by a large margin and is close to fine-tuning from the source A, which can be almost viewed as an upper bound of any algorithm's performance with AB as the source. Figure 2(b) shows that MeRLin (trained with source = AB) performs well on A, indicating it learns the transferable features. Figure 2(a) (MeRLin, train& test) corroborates the conclusion.

## 5 Theoretical Analysis with Two-layer Quadratic Neural Nets

The experiments in Section 3 demonstrate the weakness of fine-tuning and joint training. On the other hand, MeRLin is able to learn the transferable features from the source datasets. In this section, we instantiate transfer learning in a quadratic neural network where the algorithms can be rigorously studied. For a specific data distribution, we prove that (1) fine-tuning and joint training fail to learn transferable features, and (2) MeRLin recovers target ground truth with limited target examples.

**Models.** Consider a two-layer neural network $f_{\theta,\phi}(x) = g_\theta(h_\phi(x))$ with $g_\theta(z) = \theta^\top z$ and $h_\phi = \sigma(\phi^\top x)$, where $\phi = [\phi_1, \phi_2, \cdots, \phi_m] \in \mathbb{R}^{d\times m}$ is the weight of the first layer, $\theta \in \mathbb{R}^m$ is the linear head, and $\sigma(\cdot)$ is element-wise quadratic. We consider squared loss $\ell(f_{\theta,\phi}(x), y) = (f_{\theta,\phi}(x) - y)^2$.

**Source distribution.** Let $k \in \mathbb{Z}^+$ such that $2 \leq k \leq d$. We consider the following source distribution which can be solved by *multiple possible feature extractors*. Let $x_{[i]}$ denotes the $i$-th entry of $x \in \mathbb{R}^d$. Let $y = 0$ happens with prob. $1/3$, and conditioned on $y = 0$, we have $x_{[i]} = 0$ for $i \leq k$, and $x_{[i]} \sim \{\pm 1, 0\}$ uniformly randomly and independently for $i > k$. With prob. $2/3$ we have $y = 1$, and conditioned on $y = 1$, we have $x_{[i]} \sim \{\pm 1\}$ uniformly randomly and independently for $i \leq k$, and $x_{[i]} \sim \{\pm 1, 0\}$ uniformly randomly and independently for $i > k$.

The design choice here is that $x_{[1]}, \ldots, x_{[k]}$ are the useful entries for predicting the source label, because $y = x_{[i]}^2$ for any $i \leq k$. In other words, features $\sigma(e_{[i]}^\top x)$ for $i \leq k$ are useful features to learn, and any linear mixture of them works. All other entries of $x$ are independent with the label $y$.

**Target distribution.** The target distribution is exactly $k = 1$ version of the source distribution. Therefore, $y = x_{[1]}^2$, and $\sigma(e_{[1]}^\top x)$ is the correct feature extractor for the target. All other $x_{[i]}$ for $i > 1$ are independent with the label.

**Source-specific features and transferable features.** As mentioned before, $\sigma(e_{[1]}^\top x), \cdots, \sigma(e_{[k]}^\top x)$ are all good features for the source, whereas only $\sigma(e_{[1]}^\top x)$ is transferable to the target.

Since usually the source dataset is much larger than target, we assume access to *infinite* source data for simplicity, so $\widehat{\mathcal{D}}_s = \mathcal{D}_s$. We assume access to $n_t$ target data $\widehat{\mathcal{D}}_t$.

**Regularization:** Because the limited target data, the optimal solutions with unregularized objective are often not unique. Therefore, we study $\ell_2$-regularized version of the baselines and MeRLin. The regularization strength $\lambda > 0$ is selected to achieve optimal $L_{\mathcal{D}_t}$. The regularized MeRLin objective is $L_{\text{meta}}^\lambda(\theta_s, \phi) := L_{\text{meta}}(\phi, \theta_s) + \lambda(\|\theta_s\|^2 + \|\phi\|_F^2)$. The regularized joint training objective is $L_{\text{joint}}^\lambda(\theta_s, \theta_t, \phi) := L_{\text{joint}}(\theta_s, \theta_t, \phi) + \lambda(\|\theta_s\|^2 + \|\theta_t\|^2 + \|\phi\|_F^2)$. We also regularize the two objectives in the pre-training and fine-tuning. We pre-train with $L_{\mathcal{D}_s}^\lambda(\theta_s, \phi) := L_{\mathcal{D}_s}(\theta_s, \phi) + \lambda(\|\theta_s\|^2 + \|\phi\|_F^2)$, and then only fine-tune the head[2] by minimizing the target loss $L_{\widehat{\mathcal{D}}_t}^{\lambda, \hat{\phi}_{\text{pre}}}(\theta_t) := L_{\widehat{\mathcal{D}}_t}(\theta_t, \hat{\phi}_{\text{pre}}) + \lambda \|\theta_t\|^2$.

The following theorem shows that neither joint training nor fine-tuning is capable of recovering the target ground truth given limited number of target data.

**Theorem 1.** *There exists universal constants $c \in (0, 1)$ and $\epsilon > 0$, such that so long as $n_t \leq cd$, for any $\lambda > 0$, the following statements are true:*

- *With prob. at least $1 - 4\exp(-\Omega(d))$, the solution $(\hat{\theta}_s, \hat{\theta}_t, \hat{\phi}_{\text{joint}})$ of the joint training satisfies*

$$L_{\mathcal{D}_t}(\hat{\theta}_t, \hat{\phi}_{\text{joint}}) \geq \epsilon. \tag{4}$$

- *With prob. at least $1 - \frac{1}{k}$ (over the randomness of pre-training), the solution $(\hat{\theta}_t, \hat{\phi}_{\text{pre}})$ of the head-only fine-tuning satisfies*

$$L_{\mathcal{D}_t}(\hat{\theta}_t, \hat{\phi}_{\text{pre}}) \geq \epsilon. \tag{5}$$

As will be shown in the proof, not surprisingly, fine-tuning fails because it learns a random feature $\sigma(e_{[i]}^\top x)$ (where $i \in [k]$) for the source during pre-training which does not transfer to the target when $i \neq 1$. Joint training fails because it uses one neuron to learn a generalized linear model to overfit the target $n_t$ training data exactly, and then use another neuron to learn a random feature $\sigma(e_{[i]}^\top x)$ (where $i \in [k]$) for the source. The proof of Theorem 1 is deferred to Section B.

In contrast, the following theorem shows that MeRLin can recover the ground truth of the target task:

**Theorem 2.** *For any $\lambda < \lambda_0$ where $\lambda_0$ is some universal constant and any failure rate $\xi > 0$, if the target set size $n_t > \Theta(\log \frac{k}{\xi})$, with probability at least $1 - \xi$, the feature extractor $\hat{\phi}_{\text{meta}}$ found by MeRLin and the head $\hat{\theta}_t(\hat{\phi}_{\text{meta}})$ trained on $\widehat{\mathcal{D}}_t^{\text{tr}}$ recovers the ground truth of the target task:*

$$L_{\mathcal{D}_t}(\hat{\theta}_t(\hat{\phi}_{\text{meta}}), \hat{\phi}_{\text{meta}}) = 0. \tag{6}$$

Intuitively, MeRLin learns the transferable feature $\sigma(e_{[1]}^\top x)$ because it simultaneously fits the source and enables the generalization of the head on the target. The proof can be found in Section B.

---

[2]For theoretical analysis we consider only fine-tuning $\theta_t$. It is worth noting that fine-tuning both $\theta_t$ and $\phi$ converges to the same solution as target-only training, which also has large generalization gap due to overfitting.

Table 1: **Accuracy** (%) on computer vision tasks.

| Source | Fashion | SVHN | ImageNet | | | Food-101 |
|---|---|---|---|---|---|---|
| Backbone | LeNet | | ResNet-18 | | | |
| Target | USPS (600) | | CUB-200 | Caltech-256 | Stanford Cars | CUB-200 |
| Target-only | $91.07 \pm 0.45$ | $91.07 \pm 0.45$ | $32.05 \pm 0.67$ | $45.63 \pm 1.26$ | $23.22 \pm 1.02$ | $32.13 \pm 0.64$ |
| Joint training | $89.59 \pm 0.56$ | $91.54 \pm 0.32$ | $55.81 \pm 1.36$ | $78.20 \pm 0.50$ | $63.25 \pm 0.72$ | $42.08 \pm 0.59$ |
| Fine-tuning | $90.80 \pm 0.20$ | $92.12 \pm 0.39$ | $72.52 \pm 0.51$ | $81.12 \pm 0.27$ | $81.59 \pm 0.49$ | $52.30 \pm 0.51$ |
| L2-sp | $89.74 \pm 0.41$ | $91.86 \pm 0.27$ | $73.20 \pm 0.38$ | $82.31 \pm 0.22$ | $81.26 \pm 0.27$ | $53.84 \pm 0.37$ |
| **MeRLin** | $\mathbf{93.34} \pm 0.41$ | $\mathbf{93.10} \pm 0.38$ | $\mathbf{75.42} \pm 0.47$ | $\mathbf{82.45} \pm 0.26$ | $\mathbf{83.68} \pm 0.57$ | $\mathbf{58.68} \pm 0.43$ |

Table 2: **Accuracy** (%) of BERT-base on GLUE sub-tasks `dev` set.

| Target | MRPC | RTE | QNLI |
|---|---|---|---|
| Fine-tuning | $83.74 \pm 0.93$ | $68.35 \pm 0.86$ | $91.54 \pm 0.25$ |
| L2-sp | $84.31 \pm 0.37$ | $67.50 \pm 0.62$ | $91.29 \pm 0.36$ |
| **MeRLin-ft** | $\mathbf{86.03} \pm 0.25$ | $\mathbf{70.22} \pm 0.86$ | $\mathbf{92.10} \pm 0.27$ |

## 6 EXPERIMENTS

We evaluate MeRLin on several vision and NLP datasets. We show that (1) MeRLin consistently improves over baseline transfer learning algorithms including fine-tuning and joint training in both vision and NLP (Section 6.2), and (2) as indicated by our theory, MeRLin succeeds because it learns features that are more transferable than fine-tuning and joint training (Section 6.3).

### 6.1 SETUP: TASKS, MODELS, BASELINES, AND OUR ALGORITHMS

The evaluation metric for all tasks is the top-1 accuracy. We run all tasks for 3 times and report their means and standard deviations. Further experimental details are deferred to Section A.

**Datasets and models.** We consider the following four settings. The first three are object recognition problems (with different label sets). The fourth problem is the prominent NLP benchmark where the source is a language modeling task and the targets are classification problems.

*SVHN or Fashion-MNIST → USPS.* We use either SVHN (Netzer et al., 2011) (73K street view house numbers) or Fashion-MNIST Xiao et al. (2017) (50K clothes) as the source dataset. The target dataset is a random subset of 600 examples of USPS (Hull, 1994), a hand-written digit dataset. We down-sampled USPS to simulate the setting where the target dataset is much smaller than the source. We use *LeNet* (LeCun et al., 1998), a three-layer ReLU network in this experiment.

*ImageNet → CUB-200, Stanford Cars, or Caltech-256.* We use ImageNet (Russakovsky et al., 2015) as the source dataset. The target dataset is Caltech-256, CUB-200 (Wah et al., 2011), or Stanford Cars (Krause et al., 2013). These datasets have 25468, 5994, 8144 labeled examples respectively, much smaller than ImageNet with 1.2M labeled examples. Caltech is a general image classification dataset of 256 classes. Stanford Cars and CUB are fine-grained classification datasets with 196 categories of cars and 200 categories of birds, respectively. We use ResNet-18 (He et al., 2016).

*Food-101 → CUB-200.* Food (Bossard et al., 2014) is a fine-grained classification dataset of 101 classes of food. Here we validate MeRLin when the gap between the source and target is large. We also use ResNet-18.

*ImageNet → Stanford Dogs, MIT-indoors, or Aircraft.* We further test the proposed method with ResNet-50 (He et al., 2016) as baselines. We still use ImageNet (Russakovsky et al., 2015) as the source dataset. The target dataset is Stanford Dogs (Khosla et al., 2011), MIT-indoors (Quattoni & Torralba, 2009), or Aircraft (Maji et al., 2013) consisting of 12000, 5360, 8000 labeled examples, respectively.

*Language modeling → GLUE.* Pre-training on language modeling tasks and fine-tuning on labeled dataset such as GLUE (Wang et al., 2019) is dominant following the success of BERT (Devlin et al.,

Table 3: **Accuracy** (%) of computer vision tasks with ResNet-50 backbone and ImageNet.

| Target | MIT-indoors | Stanford Cars | Aircraft |
|---|---|---|---|
| Fine-tuning | $83.21 \pm 0.21$ | $84.77 \pm 0.32$ | $81.13 \pm 0.21$ |
| L2-sp | $83.98 \pm 0.29$ | $86.42 \pm 0.20$ | $80.98 \pm 0.29$ |
| DELTA | $83.66 \pm 0.20$ | $85.01 \pm 0.22$ | $80.44 \pm 0.20$ |
| BSS | $83.73 \pm 0.18$ | $86.69 \pm 0.19$ | $81.48 \pm 0.18$ |
| **MeRLin** | $\mathbf{84.50} \pm 0.26$ | $\mathbf{87.05} \pm 0.33$ | $\mathbf{82.57} \pm 0.27$ |

2019). We fine-tune BERT with MeRLin and evaluate it on the three tasks of GLUE with the smallest number of labeled examples which standard fine-tuning likely overfits.

**Baselines.** (1) *target-only*, (2) *fine-tuning*, and (3) *joint-training* have been defined in Section 2. Following standard practice, the initial learning rate of fine-tuning is $0.1\times$ the initial learning rate of pre-training to avoid overfitting. For joint training, the overall objective can be formulated as: $(1 - \alpha)L_{\widehat{\mathcal{D}}_s} + \alpha L_{\widehat{\mathcal{D}}_t}$. We tune $\alpha$ to achieve optimal performance. The fourth baseline is (4) *L2-sp* (Li et al., 2018), which fine-tunes the models with a regularization penalizing the parameter distance to the pre-trained feature extractor. (5) DELTA (Li et al., 2019) constrains the layer outputs selected with attention. (6) BSS (Chen et al., 2019) regularized the singular value of feature matrix. We also tuned the strength the L2-sp, Delta and BSS regularizer.

**MeRLin.** We perform standard training with cross entropy loss on the source domain while meta-learning the representation in the target domain as described in Section 4.

**MeRLin-ft.** In BERT experiments, training on the source masked language modeling task is prohibitively time-consuming, so we opt to a light-weight variant instead: start from pre-trained BERT, and only meta-learn the representation in the target domain.

## 6.2 RESULTS

Results of digits classification and object recognition are provided in Table 1. MeRLin consistently outperforms all baselines. Note that the discrepancy between Fashion-MNIST and USPS is very large, where fine-tuning and joint training perform even worse than target-only. Nonetheless, MeRLin is still capable of harnessing the knowledge from the source domain. On Food-101→CUB-200, MeRLin improves over fine-tuning by $6.58\%$, indicating that MeRLin helps learn transferable features even when the gap between the source and target tasks is huge. In Table 2, we validate our method on GLUE tasks. MeRLin-ft outperforms standard BERT fine-tuning and L2-sp. Since MeRLin-ft only changes the training objective of fine-tuning, it can be easily applied to NLP models.

## 6.3 ANALYSIS

We empirically analyze the representations and verify that MeRLin indeed learns more transferable features than fine-tuning and joint training.

**Intra-class to inter-class variance ratio.** Suppose the representation of the j-th example of the i-th class is $\phi_{i,j}$. $\mu_i = \frac{1}{N_i}\sum_{j=1}^{N_i}\phi_{i,j}$, and $\mu = \frac{1}{C}\sum_{i=1}^{C}\mu_i$. Then the intra-class to inter-class variance ratio can be calculated as $\frac{\sigma_{intra}^2}{\sigma_{inner}^2} = \frac{C}{N}\frac{\sum_{i,j}\|\phi_{i,j}-\mu_i\|^2}{\sum_i\|\mu_i-\mu\|^2}$. Low values of this ratio correspond to representations where classes are well-separated. Results on ImageNet $\to$ CUB-200 and Stanford Cars task are shown in Figure 3. MeRLin reaches much smaller ratio than baselines.

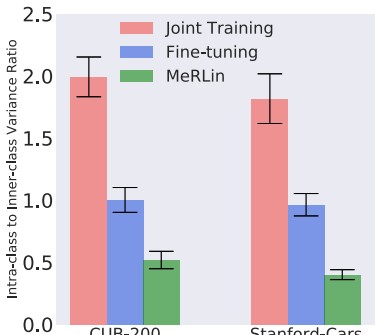

Figure 3: **Comparison of intra-class to inter-class variance ratio.** This quantity is lowest for MeRLin, indicating that it separates classes best.

## 7 ADDITIONAL RELATED WORK

Transfer learning is prevalent in deep learning applications. In computer vision, ImageNet pre-training is a common practice for nearly all target tasks. Early works (Oquab et al., 2014; Donahue et al., 2014) directly apply ImageNet features to target tasks. Fine-tuning from ImageNet pre-trained models has become dominant (Long et al., 2015; He et al., 2017; Kolesnikov et al., 2019). On the other

hand, transfer learning is also crucial to the success of NLP. Pre-training transformers on large-scale language tasks boosts performance on downstream tasks (Devlin et al., 2019; Yang et al., 2019).

A recent line of literature casts doubt on the consistency of transfer learning's success (Raghu et al., 2019; He et al., 2018; Kornblith et al., 2019). Huh et al. (2016) observed that some set of examples in ImageNet are more transferable than the others. Geirhos et al. (2019) found out that the texture of ImageNet is not transferable to some target tasks. Training on the source dataset may also need early stopping to find optimal transferability Liu et al. (2019a); Neyshabur et al. (2020).

Meta-learning, originated from the *learning to learn* idea (Hochreiter et al., 2001; Vilalta & Drissi, 2002; Maclaurin et al., 2015; Zoph & Le, 2016), learns from multiple training tasks models that can be swiftly adapted to new tasks (Finn et al., 2017; Rajeswaran et al., 2019; Nichol et al., 2018b). Raghu et al. (2020); Goldblum et al. (2020) empirically studied the mechanism of MAML's success. Computationally, our method uses bi-level optimization techniques similar to meta-learning work. E.g., Bertinetto et al. (2019) speeds up the implementation of MAML Finn et al. (2017) with closed-form solution of the inner loop, which is a technique that we also use. However, the key difference between our paper from the meta-learning approach is that we only learn from a single target task and evaluate on it. Therefore, conceptually, our algorithm is closer to the learning-to-learn approach for hyperparameter optimization (Maclaurin et al., 2015; Zoph & Le, 2016), where there is a single distribution that generates the training and validation dataset.

## 8 CONCLUSION

We study the limitations of fine-tuning and joint training. To overcome their drawbacks, we propose meta representation learning to learn transferable features. Both theoretical and empirical evidence verify our findings. Results on vision and NLP tasks validate our method on real-world datasets. Our work raises many intriguing questions for further study. Could we apply meta-learning to heterogeneous target tasks? What's more, future work can pay attention to disentangling transferable features from non-transferable features explicitly for better transfer learning.

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

## A    ADDITIONAL DETAILS OF EXPERIMENTS

### A.1    THE SEMI-SYNTHETIC EXPERIMENT

The original CIFAR images are of resolution $32 \times 32$. For the transferable dataset A, we reserve the upper $16 \times 32$ and fill the lower half with $[0.485, 0.456, 0.406]$ for the three channels (the mean of CIFAR-10 images). For the non-transferable dataset B, the lower part $16 \times 32$ pixels are generated with i.i.d. gaussian distribution with the upper half filled with $[0.485, 0.456, 0.406]$ similarly. To make the non-transferable part related to the labels, we set the mean of the gaussian distribution to $0.1 \times c$, where $c$ is the class index of the image. The variance of the gaussian noise is set to $0.2$. We always clamp the images to $[0, 1]$ to make the generated images valid. For the source dataset, we use $49500$ CIFAR-10 images, while for the target, we use the other $500$ to avoid memorizing target examples.

We use ResNet-32 implementation provided in `github.com/akamaster/pytorch_resnet_cifar10`. We set the initial learning rate to $0.1$, and decay the learning rate by $0.1$ after every 50 epochs. We use t-SNE (van der Maaten & Hinton, 2008) visualizations provided in `sklearn`. The perplexity is set to 80.

### A.2    IMPLEMENTATION ON REAL DATASETS

We implement all models on PyTorch with 2080Ti GPUs. All models are optimized by SGD with 0.9 momentum. For object recognition tasks, the initial learning rate is set to 0.01, with $5 \times 10^{-4}$ weight decay. The batch-size is set to 64. We run each model for 100 epochs. ImageNet pre-trained models can be found in `torchvision`. We use a batch size of 128 on the source dataset and 512 on the target dataset. The initial learning rate is set to 0.1 for training from scratch and 0.01 for ImageNet initialization. We decay the learning rate by 0.1 every 50 epochs until 150 epochs. The weight decay is set to $5 \times 10^{-4}$. For GLUE tasks, we follow the standard practice of Devlin et al. (2019). The BERT model is provided in `github.com/Meelfy/pytorch_pretrained_BERT`. For each model, we set the head (classifier) to the top one linear layer. We use a batch size of 32. The learning rate is set to $5 \times 10^{-5}$ with 0.1 warmup proportion. During fine-tuning, the initial learning rate is 10 times smaller than training from scratch following standard practice. The hyper-parameter $\rho$ is set to 2, and $\lambda$ is found with cross validation. We randomly split the training set into 80% and 20%. We train on the 80% part and use the rest 20% as the validation set. We also provide the results of varying $\rho$ and $\lambda$ in Section A.7. In object recognition however, we also provides a pre-trained version: starting from ImageNet pre-trained solution.

### A.3    IMPLEMENTING THE SPEED-UP VERSION

**Practical implementation: speeding up with MSE loss.** Training the head $g_\theta$ in the inner loop of meta learning can be time-consuming. Even using implicit function theorem or implicit gradients as proposed in (Bai et al., 2019; Rajeswaran et al., 2019; Lin et al., 2020), we have to approximate the inverse of Hessian. To solve the optimization issues, we propose to analytically calculate the prediction of the linear head $\theta_t$ and directly back-prop to the feature extractor $h_\phi$. Thus, we only need to compute the gradient once in a single step. Concretely, suppose we use MSE-loss. Denote by $\mathbf{H} \in \mathbb{R}^{\frac{n_t}{2} \times m}$ the feature matrix of the $\frac{n_t}{2}$ target samples in the target meta-training set $\widehat{\mathcal{D}}_t^{\mathrm{tr}}$. Then $\widetilde{\theta}_t$ in equation 1 can be analytically computed as $\widehat{\theta}_t = (\mathbf{H}\mathbf{H}^\top + \lambda\mathbf{I})^{-1}\mathbf{y}$, where $\lambda$ is a hyper-parameter for regularization. The objective of the outer loop can be directly computed as

$$\underset{\phi \in \boldsymbol{\Phi}, \theta_s \in \boldsymbol{\Theta}}{\text{minimize}} \; J(\phi, \theta_s) = L_{\widehat{\mathcal{D}}_s}(\theta_s, \phi) + \rho \frac{2}{n_t} \sum_{i=1}^{\frac{n_t}{2}} \ell(g_{(\mathbf{H}\mathbf{H}^\top + \lambda\mathbf{I})^{-1}\mathbf{y}} \circ h_\phi(x_i^{t'}), y_i^{t'}). \qquad (7)$$

We implement the kernel speed-up version on classification tasks following Arora et al. (2019b). We treat the classification problems as multi-variate ridge regression. Suppose we have label $c \in \{1, 2, \cdots, 10\}$. Then the target encoding for regression is $-0.1 \times \mathbf{1} + \mathbf{e}_c$. For example, if the label is 3, then the encoding will be $(-0.1, -0.1, 0.9, \cdots, -0.1)$. Then the parameters of the target head in the inner loop can be computed as $\widehat{\theta}_t = (\mathbf{H}\mathbf{H}^\top + \lambda\mathbf{I})^{-1}\mathbf{Y}$. We then compute the MSE loss

on the target validation set: $\|\mathbf{H}_{val}\widehat{\theta}_t - \mathbf{Y}_{val}\|_2^2$ in the outer loop. We summarize the details of the vanilla version and the speed-up version in Algorithm 1 and Algorithm 2.

---

**Algorithm 1** Meta Representation Learning (MeRLin).

---

1: **Input:** the source dataset $\widehat{\mathcal{D}}_s$ and the evolving target dataset $\widehat{\mathcal{D}}_t$.
2: **Output:** learned representations $\phi$.
3: **for** iter $= 0$ **to** MaxIter **do**
4:     Initialize the target head $\theta_t$.
5:     Randomly sample target train set $\widehat{\mathcal{D}}_t^{\text{tr}}$ and target validation set $\widehat{\mathcal{D}}_t^{\text{val}}$ from $\widehat{\mathcal{D}}_t$.
6:     **for** $t = 0$ **to** n **do**
7:         Train the target head on $\widehat{\mathcal{D}}_t^{\text{tr}}$:
$$\theta_t \leftarrow \theta_t - \eta \nabla_{\theta_t} L_{\widehat{\mathcal{D}}_t^{\text{tr}}}(\theta_t, \phi).$$
8:     **end for**
9:     In the outer loop, update the representation $\phi$ and the source head $\theta_s$:
$$(\phi, \theta_s) \leftarrow (\phi, \theta_s) - \eta \nabla_{(\phi, \theta_s)} \left[ L_{\widehat{\mathcal{D}}_s}(\theta_s, \phi) + \rho L_{\widehat{\mathcal{D}}_t^{\text{val}}}(\theta_t, \phi) \right].$$
10: **end for**

---

**Algorithm 2** Meta Representation Learning (MeRLin): speed-up implementation.

---

1: **Input:** the source dataset $\widehat{\mathcal{D}}_s$ and the evolving target dataset $\widehat{\mathcal{D}}_t$.
2: **Output:** learned representations $\phi$.
3: **for** $t = 0$ **to** MaxIter **do**
4:     Randomly sample target train set $\widehat{\mathcal{D}}_t^{\text{tr}}$ and target validation set $\widehat{\mathcal{D}}_t^{\text{val}}$ from $\widehat{\mathcal{D}}_t$.
5:     Analytically calculate the solution of target head $\widehat{\theta}_t(\phi)$ in the inner loop
$$\widehat{\theta}_t(\phi) = (\mathbf{H}\mathbf{H}^\top + \lambda \mathbf{I})^{-1}\mathbf{Y}.$$
6:     In the outer loop, update the representation $\phi$ and the source head $\theta_s$:
$$(\phi, \theta_s) \leftarrow (\phi, \theta_s) - \eta \nabla_{(\phi, \theta_s)} \left[ L_{\widehat{\mathcal{D}}_s}(\theta_s, \phi) + \rho \frac{2}{n_t} \|\mathbf{H}_{val}\widehat{\theta}_t - \mathbf{Y}_{val}\|_2^2 \right].$$
7: **end for**

---

## A.4 DATASETS

**CUB-200** (Wah et al., 2011) is a fine-grained dataset of 200 bird species. The training dataset consists of 5994 images and the test set consists of 5794 images. `http://www.vision.caltech.edu/visipedia/CUB-200-2011.html`

**Stanford Cars** (Krause et al., 2013) dataset contains 16,185 images of 196 classes of cars. The data is split into 8,144 training images and 8,041 testing images. `http://ai.stanford.edu/~jkrause/cars/car_dataset.html`

**Food-101** (Bossard et al., 2014) is a fine-grained classification dataset of 101 kinds of food, with 750 training images and 250 test images for each kind. `http://www.vision.ee.ethz.ch/datasets_extra/food-101/`

**Caltech-256** (Griffin et al., 2007) is an object recognition dataset of 256 categories. In our experiments, the training set consists of 25468 images, and the test set consists of 5139 images.`http://www.vision.caltech.edu/Image_Datasets/Caltech256/`

**MIT-indoors** (Quattoni & Torralba, 2009) is a scene recognition dataset of 67 classes. The training set contains 80 examples for each class, and the test set contains 20 examples for each class. `http://web.mit.edu/torralba/www/indoor.html`

**Stanford-Cars** (Khosla et al., 2011) is a fine-grained classification dataset of 120 kinds of dogs. `http://vision.stanford.edu/aditya86/ImageNetDogs/main.html`

**Aircraft** (Maji et al., 2013) is an object recognition dataset of 100 aircraft variants. Each class has 80 images for training and 20 images for testing. `http://www.robots.ox.ac.uk/~vgg/data/fgvc-aircraft/`

**MNIST** (LeCun et al., 1998) is a dataset of hand-written digits. It has a training set of 60,000 examples, and a test set of 10,000 examples. We randomly sample 600 samples as the training set for the target domain. http://yann.lecun.com/exdb/mnist/

**SVHN** (Netzer et al., 2011) is a real-world image dataset of street view house numbers. It has 73257 digits for training, 26032 digits for testing. In Figure 3 of the main text, we randomly sample 600 digits for training, and all the 26032 digits are used in testing. http://ufldl.stanford.edu/housenumbers/

**USPS** (Hull, 1994) is a database for handwritten text recognition. It has 7291 train and 2007 test images. We randomly sample 600 samples as the training set for the target domain. https://www.kaggle.com/bistaumanga/usps-dataset

### A.5 FURTHER ABLATION STUDY.

We extend the last column of Table 2 in Table 4. We further compare with two variants of MeRLin as ablation study:

**MeRLin (pre-trained).** We first pre-train the model on the source dataset and then optimize the MeRLin objective starting from the pre-trained solution.

**MeRLin-target-only.** MeRLin-target-only only meta-learns representations on the target domain starting from random initialization. We test whether the meta-learning objective itself has regularization effect.

Table 4: Accuracy on Food → CUB.

| Algorithm | Target-only | Fine-tuning | Joint Training | MeRLin-target-only | MeRLin (pre-trained) | MeRLin |
|---|---|---|---|---|---|---|
| Accuracy | $32.10 \pm 0.64$ | $52.30 \pm 0.51$ | $42.08 \pm 0.59$ | $40.17 \pm 0.44$ | $55.26 \pm 0.43$ | $58.68 \pm 0.43$ |

MeRLin (pre-trained) performs worse than MeRLin, but it still improves over fine-tuning and joint training. Note-that MeRLin (pre-trained) only need to train on ImageNet for 2 epochs, much shorter than joint training. MeRLin-target-only improves target-only by $8\%$, indicating that meta-learning helps avoid overfitting even without the source dataset.

### A.6 FEATURE-LABEL CORRELATION

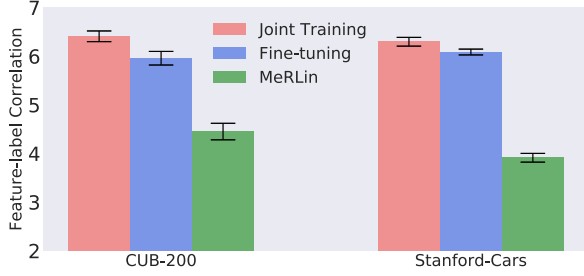
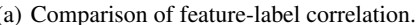
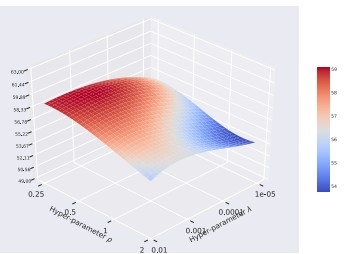

(a) Comparison of feature-label correlation.

(b) Sensitivity to hyper-parameters.

Figure 4: (a) **Analysis of Feature Quality.** Comparison of feature-label correlation. A lower quantity is better, and MeRLin has the lowest value. (b) **Sensitivity of the proposed method to hyper-parameters.** We test the accuracy on Food-101→CUB-200 with varying $\rho$ and $\lambda$ and provide the visualization.

Suppose the feature matrix is $\mathbf{H}$, and the label vector is $\mathbf{y}$, then the correlation between feature and label can be defined as $\mathbf{y}^\top \left(\mathbf{H}^\top \mathbf{H}\right)^{-1} \mathbf{y}$. As is shown by Arora et al. (2019a); Cao & Gu (2019), this term is closely related to the generalization error of neural networks, with a smaller quantity indicating better generalization. We calculate $\mathbf{y}^\top \left(\mathbf{H}^\top \mathbf{H}\right)^{-1} \mathbf{y}$ on ImageNet → CUB-200 and Stanford Cars. As shown in Figure 4(a), the features learned by MeRLin are more closely related

to labels than fine-tuning and joint training, indicating MeRLin is indeed learning more transferable features compared with baselines.

## A.7 SENSITIVITY OF THE PROPOSED METHOD TO HYPER-PARAMETERS.

We test the model on Food-101$\rightarrow$CUB-200 with varying hyper-parameters $\rho$ and $\lambda$. Results in Figure 4(b) indicate that the model is not sensitive to varying $\rho$ and $\lambda$. Intuitively, larger $\rho$ indicates more emphasis on the target meta-task. When $\rho$ approaches 0, the performance of MRL is approaching fine-tuning. $\lambda$ exerts regularization to the classifier in the inner loop training. It is also note worthy that $\lambda$ can avoid the problem that $\mathbf{H}\mathbf{H}^{\top}$ is occasionally invertible. Without $\lambda$ the model can fail to converge sometimes.

## A.8 BOOTSTRAPPED CONFIDENCE INTERVAL OF OBJECTION RECOGNITION RESULTS

We run 10 seeds of each experiment on L2-sp and MeRLin in Table 1 and calculate 95% bootstrapped confidence interval of each results in Figure 5.

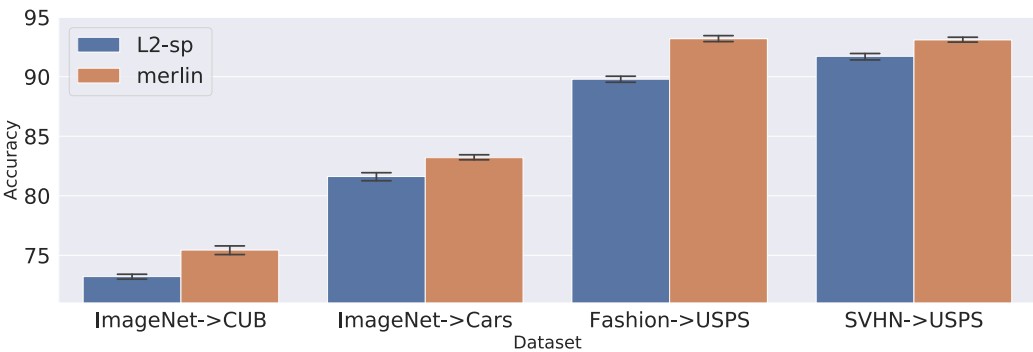

Figure 5: **Comparison of L2-sp and MeRLin**. The results are averaged over 10 runs, with error bars representing 95% confidence interval drawn by 1,000 bootstraps.

# B    MISSING DETAILS IN SECTION 5

## B.1    PROOF OF THEOREM 1

**Lemma 1.** *Suppose $0 \leq \epsilon \leq \frac{2}{3}$. For each solution $\theta_s, \theta_t, \phi$ satisfying $\mathbb{E}_{x,y \sim \mathcal{D}_t} [\ell(f_{\theta_t,\phi}(x), y)] \leq \epsilon$, the joint training objective $L_{\text{joint}}^{\lambda}(\theta_s, \theta_t, \phi)$ is lower bounded:*

$$L_{\text{joint}}^{\lambda}(\theta_s, \theta_t, \phi) = (1-\alpha)L_{\mathcal{D}_s}(\theta_s, \phi) + \alpha L_{\widehat{\mathcal{D}}_t}(\theta_t, \phi) + \lambda \left( \|\theta_s\|^2 + \|\theta_t\|^2 + \|\phi\|_F^2 \right)$$

$$\geq \min_{\mu} \left( \frac{3\lambda}{2^{4/3}} |\mu|^{2/3} + \frac{2}{3}(1-\alpha)(\mu-1)^2 \right) + \frac{3\lambda}{2^{4/3}} \left( 1 - \sqrt{\frac{3\epsilon}{2}} \right)^{2/3}.$$

*Proof of Lemma 1.*  Define $d \times d$ matrix

$$A = \sum_{i=1}^{m} \theta_{si}\phi_i\phi_i^{\top}. \tag{8}$$

Define $x_{[1:k]}$ and $x_{[k+1:d]}$ be the first $k$ and last $d-k$ dimensions of $x$. $A_{k,k}$, $A_{k,\bar{k}}$ and $A_{\bar{k},\bar{k}}$ be $k \times k$, $k \times (d-k)$ and $(d-k) \times (d-k)$ matrices that correspond to the upper left, upper right and lower right part of $A$. For a random vector $x$ where the first $k$ dimensions are uniformly independently from $\{\pm 1\}$, the last $d-k$ dimensions are uniformly indepdently from $\{0, \pm 1\}$, define random variables $A_1 = x_{[1:k]}^{\top} A_{k,k} x_{[1:k]}$, $A_2 = x_{[k+1:d]}^{\top} A_{\bar{k},\bar{k}} x_{[k+1:d]}$. (Note that $x$ is defined on a different distribution than $\mathcal{D}_s$.)

We have bound

$$(1-\alpha)\mathbb{E}_{x,y \sim \mathcal{D}_s} [\ell(f_{\theta_s,\phi}(x), y)] + \lambda \left( \|\theta_s\|^2 + \frac{1}{2}\|\phi\|_F^2 \right) \tag{9}$$

$$=(1-\alpha)\mathbb{E}_{x,y \sim \mathcal{D}_s} \left[ \left( x_{[1:k]}^{\top} A_{k,k} x_{[1:k]} + 2x_{[1:k]}^{\top} A_{k,\bar{k}} x_{[k+1:d]} + x_{[k+1:d]}^{\top} A_{\bar{k},\bar{k}} x_{[k+1:d]} - y \right)^2 \right]$$

$$+ \lambda \left( \|\theta_s\|^2 + \frac{1}{2}\|\phi\|_F^2 \right) \tag{10}$$

$$\geq(1-\alpha)\mathbb{E}_{x,y \sim \mathcal{D}_s} \left[ \left( x_{[1:k]}^{\top} A_{k,k} x_{[1:k]} + x_{[k+1:d]}^{\top} A_{\bar{k},\bar{k}} x_{[k+1:d]} - y \right)^2 \right] + \lambda \left( \|\theta_s\|^2 + \frac{1}{2}\|\phi\|_F^2 \right) \tag{11}$$

$$=(1-\alpha) \left( \frac{2}{3}\mathbb{E} \left[ (A_1-1)^2 \right] + \frac{4}{3}\mathbb{E} \left[ (A_1-1)A_2 \right] + \mathbb{E} \left[ A_2^2 \right] \right) + \lambda \left( \|\theta_s\|^2 + \frac{1}{2}\|\phi\|_F^2 \right) \tag{12}$$

$$=(1-\alpha) \left( \frac{2}{3}\mathbb{E} \left[ (A_1+A_2-1)^2 \right] + \frac{1}{3}\mathbb{E} \left[ A_2^2 \right] \right) + \lambda \left( \|\theta_s\|^2 + \frac{1}{2}\|\phi\|_F^2 \right) \tag{13}$$

$$\geq\frac{2}{3}(1-\alpha) \left( \mathbb{E} \left[ A_1 + A_2 \right] - 1 \right)^2 + \frac{3\lambda}{2^{4/3}} \left( |\mathbb{E} \left[ A_1 + A_2 \right] | \right)^{2/3} \tag{14}$$

The first inequality is because

$$\mathbb{E}_{x,y \sim \mathcal{D}_s} \left[ \left( x_{[1:k]}^{\top} A_{k,k} x_{[1:k]} \right) \left( x_{[1:k]}^{\top} A_{k,\bar{k}} x_{[k+1:d]} \right) \right] = 0, \tag{15}$$

$$\mathbb{E}_{x,y \sim \mathcal{D}_s} \left[ \left( x_{[k+1:d]}^{\top} A_{\bar{k},\bar{k}} x_{[k+1:d]} \right) \left( x_{[1:k]}^{\top} A_{k,\bar{k}} x_{[k+1:d]} \right) \right] = 0, \tag{16}$$

$$\mathbb{E}_{x,y \sim \mathcal{D}_s} \left[ \left( x_{[1:k]}^{\top} A_{k,\bar{k}} x_{[k+1:d]} \right) y \right] = 0. \tag{17}$$

The second inequality is because

$$\sum_{i=1}^{m} \left( (\theta_{si})^2 + \frac{1}{2} \|\phi_i\|^2 \right) \geq \frac{3}{2^{4/3}} \sum_{i=1}^{m} \left( |\theta_{si}| \cdot \|\phi_i\|^2 \right)^{2/3} \tag{18}$$

$$\geq \frac{3}{2^{4/3}} \left( \sum_{i=1}^{m} |\theta_{si}| \cdot \|\phi_i\|^2 \right)^{2/3} \tag{19}$$

$$\geq \frac{3}{2^{4/3}} \left( \sum_{i=1}^{m} |A_{[i,i]}| \right)^{2/3}, \tag{20}$$

where the first inequality is AM-GM inequality, the second inequality is by concavity of $(\cdot)^{2/3}$. The third inequality is because for diagonal matrix $D$ that has 1 at $(i,i)$ if $A_{[i,i]} \geq 0$, $-1$ at $(i,i)$ if $A_{[i,i]} < 0$, we have

$$\sum_{i=1}^{m} |A_{[i,i]}| = tr(AD) = \sum_{i=1}^{m} \theta_{si} \phi_i^\top D \phi_i \leq \sum_{i=1}^{m} |\theta_{si}| \cdot \|\phi_i\|^2. \tag{21}$$

On the other hand, for the target, we define $d \times d$ matrix

$$B = \sum_{i=1}^{m} \theta_{ti} \phi_i \phi_i^\top. \tag{22}$$

Define $x_{[1]}$ and $x_{[2:d]}$ be the first 1 and last $d-1$ dimensions of $x$. $B_{1,1}$, $B_{1,\bar{1}}$ and $B_{\bar{1},\bar{1}}$ be $1 \times 1$, $1 \times (d-1)$ and $(d-1) \times (d-1)$ matrices that correspond to the upper left, upper right and lower right part of $B$. For a random vector $x$ where the first dimension is uniformly independently from $\{\pm 1\}$, the last $d-1$ dimensions are uniformly indepdently from $\{0, \pm 1\}$, define random variables $B_1 = x_{[1]}^\top B_{1,1} x_{[1]}$, $B_2 = x_{[2:d]}^\top B_{\bar{1},\bar{1}} x_{[2:d]}$. (Note that $x$ is defined on a different distribution than $\mathcal{D}_t$.)

Using similar argument as above, we have

$$\mathbb{E}_{x,y \sim \mathcal{D}_t} \left[ \ell(f_{\theta_t,\phi}(x), y) \right] \geq \frac{2}{3} \left( \mathbb{E} \left[ B_1 + B_2 \right] - 1 \right)^2. \tag{23}$$

However, we know that $\mathbb{E}_{x,y \sim \mathcal{D}_t} \left[ \ell(f_{\theta_t,\phi}(x), y) \right] \leq \epsilon$, so there has to be

$$\mathbb{E} \left[ B_1 + B_2 \right] \geq 1 - \sqrt{\frac{3\epsilon}{2}}, \tag{24}$$

therefore we have

$$\lambda \left( \|\theta_t\|^2 + \frac{1}{2} \|\phi\|_F^2 \right) \geq \frac{3\lambda}{2^{4/3}} \left( 1 - \sqrt{\frac{3\epsilon}{2}} \right)^{2/3}. \tag{25}$$

Summing up Equation 9 and Equation 25 finishes the proof.

$\square$

**Lemma 2.** *Assume $\hat{\phi}_t$ is a vector such that $\langle \hat{\phi}_t, x_i^t \rangle = x_{i[1]}^t$ for all $x_i^t \in \widehat{\mathcal{D}}_t$, then there exists some solution $(\theta_s, \theta_t, \phi)$ such that*

$$L_{\text{joint}}^{\lambda}(\theta_s, \theta_t, \phi) \leq \min_{\mu} \left( \frac{3\lambda}{2^{2/3}} |\mu|^{2/3} + \frac{2}{3}(1-\alpha)(\mu-1)^2 \right) + \frac{3\lambda}{2^{2/3}} \left\| \hat{\phi}_t \right\|_2^{4/3}. \tag{26}$$

*Proof of Lemma 2.* Assume $\mu^* \in \arg\min_{\mu} \left( \frac{3\lambda}{2^{2/3}} |\mu|^{2/3} + \frac{2}{3}(1-\alpha)(\mu-1)^2 \right)$, then obviously $\mu^* \in [0,1]$. Let $\phi_1 = (\sqrt{2}\mu^*)^{1/3} e_1$, $\phi_2 = \frac{2^{1/6}}{\|\hat{\phi}_t\|_2^{1/3}} \hat{\phi}_t$, $\phi_i = 0$ for $i > 2$, $\theta_s = (\mu^*/2)^{1/3} e_1$ and $\theta_t = \frac{\|\hat{\phi}_t\|_2^{2/3}}{2^{1/3}} e_2$. Now we prove that this model satisfies the Equation 26.

First of all, we notice that for any $x_i^t \in \widehat{\mathcal{D}}_t$, there is

$$x_i^{t\top} \left( \sum_{i=1}^m \theta_{ti} \phi_i \phi_i^\top \right) x_i^t \tag{27}$$

$$= x_i^{t\top} \theta_{t2} \phi_2 \phi_2^\top x_i^t \tag{28}$$

$$= \langle \hat{\phi}_t, x_i^t \rangle^2 \tag{29}$$

$$= y_i^t. \tag{30}$$

Therefore we have

$$L_{\widehat{\mathcal{D}}_t}(\theta_t, \phi) = 0 \tag{31}$$

.

On the other hand, we have

$$(1-\alpha)L_{\mathcal{D}_s}(\theta_s, \phi) + \lambda \left( \|\theta_s\|^2 + \|\phi_1\|^2 \right) \tag{32}$$

$$= \frac{2}{3}(1-\alpha)(\mu^* - 1)^2 + \frac{3\lambda}{2^{2/3}} |\mu^*|^{2/3} \tag{33}$$

$$= \min_\mu \left( \frac{3\lambda}{2^{2/3}} |\mu|^{2/3} + \frac{2}{3}(1-\alpha)(\mu - 1)^2 \right). \tag{34}$$

Plugging Equation 31 and Equation 32 into the formula of $L_{\text{joint}}^\lambda(\theta_s, \theta_t, \phi)$ finishes the proof. $\qquad\square$

**Lemma 3.** *Let $X \in \mathbb{R}^{d \times n}$ be a random matrix where each entry is uniformly random and independently sample from $\{0, \pm 1\}$, $n < \frac{d}{2}$. Let $P_X e_1$ be the projection of $e_1$ to the column space of $X$. Then, there exists absolute constants $c_0 > 0$ and $C > 0$, such that with probability at least $1 - 4\exp(-Cd)$, there is*

$$\|P_X e_1\|_2 \leq c_0 \sqrt{\frac{n}{d}}. \tag{35}$$

*Proof of Lemma 3.* Let $s_{min}(X)$ and $s_{max}(X)$ be the minimal and maximal singular values of $X$ respectively. Then we have

$$\|P_X e_1\|_2 = \left\| X(X^\top X)^{-1} X^\top e_1 \right\|_2 \tag{36}$$

$$\leq \|X\|_{op} \left\| (X^\top X)^{-1} \right\|_{op} \left\| X^\top e_1 \right\|_2 \tag{37}$$

$$\leq s_{max}(X)(s_{min}(X))^{-2} \sqrt{n}. \tag{38}$$

By Theorem 3.3 in Rudelson & Vershynin (2010), there exists constants $c_1, c_2 > 0$, such that

$$P(s_{min}(X) \leq c_1(\sqrt{d} - \sqrt{n})) \leq 2\exp(-c_2 d). \tag{39}$$

By Proposition 2.4 in Rudelson & Vershynin (2010), there exists constants $c_3, c_4 > 0$, such that

$$P(s_{max}(X) \geq c_3(\sqrt{d} + \sqrt{n})) \leq 2\exp(-c_4 d). \tag{40}$$

Let $C = min\{c_2, c_4\}$, then with probability at least $1 - 4\exp(Cd)$, there is

$$s_{max}(X)(s_{min}(X))^{-2}\sqrt{n} \tag{41}$$

$$\leq \frac{c_3}{c_1^2} \frac{\sqrt{d} + \sqrt{n}}{(\sqrt{d} - \sqrt{n})^2} \sqrt{n} \tag{42}$$

$$\leq \frac{(2 + \sqrt{2})c_3}{(\sqrt{2} - 1)^2 c_1^2} \sqrt{\frac{n}{d}}, \tag{43}$$

which completes the proof.

$\qquad\square$

*Proof of Theorem 1.* We prove the joint training part of Theorem 1 following this intuition: (1) the total loss of each solution with target loss $\mathbb{E}_{x,y\sim\mathcal{D}_t}\left[\ell(f_{\theta_t,\phi}(x),y)\right] \leq \epsilon$ is lower bounded as indicated by Lemma 1, and (2) there exists a solution with loss smaller than the aforementioned lower bound as indicated by Lemma 2.

By Lemma 1, for any $\theta_s, \theta_t, \phi$ satisfying $\mathbb{E}_{x,y\sim\mathcal{D}_t}\left[\ell(f_{\theta_t,\phi}(x),y)\right] \leq \epsilon$, the joint training loss $L_{\text{joint}}^\lambda(\theta_s, \theta_t, \phi)$ is lower bounded,

$$L_{\text{joint}}^\lambda(\theta_s, \theta_t, \phi) \geq \min_\mu \left(\frac{3\lambda}{2^{4/3}}|\mu|^{2/3} + \frac{2}{3}(1-\alpha)(\mu-1)^2\right) + \frac{3\lambda}{2^{4/3}}\left(1-\sqrt{\frac{3\epsilon}{2}}\right)^{2/3}. \tag{44}$$

Let $P_X e_1$ be the projection of vector $e_1$ to the subspace spanned by the target data. According to Lemma 2, there exists some solution $(\theta_s, \theta_t, \phi)$ such that

$$L_{\text{joint}}^\lambda(\theta_s, \theta_t, \phi) \leq \mu\left(\frac{3\lambda}{2^{2/3}}|\mu|^{2/3} + \frac{2}{3}(1-\alpha)(\mu-1)^2\right) + \frac{3\lambda}{2^{2/3}}\left\|\hat{\phi}_t\right\|_2^{4/3}. \tag{45}$$

Let $\epsilon_0 > 0$ be a constant such that $\frac{1}{2^{4/3}}\left(1-\sqrt{\frac{3\epsilon_0}{2}}\right)^{2/3} > \frac{1}{2^{2/3}} - \frac{1}{2^{4/3}}$. According to Lemma 3, there exists absolute constants $c \in (0,1), C > 0$, such that so long as $n_t \leq cd$, there is with probability at least $1 - 4\exp(-Cd)$,

$$\frac{1}{2^{4/3}}\left(1-\sqrt{\frac{3\epsilon_0}{2}}\right)^{2/3} > \frac{1}{2^{2/3}} - \frac{1}{2^{4/3}} + \frac{1}{2^{2/3}}\left\|\hat{\phi}_t\right\|_2^{4/3}. \tag{46}$$

Now we prove the upper bound in Equation 45 is smaller than the lower bound in Equation 44. This is because

$$\min_\mu \left(\frac{3\lambda}{2^{2/3}}|\mu|^{2/3} + \frac{2}{3}(1-\alpha)(\mu-1)^2\right) + \frac{3\lambda}{2^{2/3}}\left\|\hat{\phi}_t\right\|_2^{4/3} \tag{47}$$

$$= \min_\mu \left(\frac{3\lambda}{2^{2/3}}|\mu|^{2/3} + \frac{2}{3}(1-\alpha)(\mu-1)^2\right) - 3\lambda\left(\frac{2^{2/3}-1}{2^{4/3}}\right) + \frac{3\lambda}{2^{2/3}}\left\|\hat{\phi}_t\right\|_2^{4/3} + 3\lambda\left(\frac{1}{2^{2/3}} - \frac{1}{2^{4/3}}\right) \tag{48}$$

$$\leq \min_\mu \left(\frac{3\lambda}{2^{4/3}}|\mu|^{2/3} + \frac{2}{3}(1-\alpha)(\mu-1)^2\right) + \frac{3\lambda}{2^{2/3}}\left\|\hat{\phi}_t\right\|_2^{4/3} + 3\lambda\left(\frac{1}{2^{2/3}} - \frac{1}{2^{4/3}}\right) \tag{49}$$

$$\leq \min_\mu \left(\frac{3\lambda}{2^{4/3}}|\mu|^{2/3} + \frac{2}{3}(1-\alpha)(\mu-1)^2\right) + \frac{3\lambda}{2^{4/3}}\left(1-\sqrt{\frac{3\epsilon_0}{2}}\right)^{2/3}, \tag{50}$$

where the first inequality uses that fact that $|\mu| < 1$ for the optimal $\mu$, the second inequality is by Equation 46. This completes the proof for joint training.

Then, we prove the result about fine-tuning. According to Lemma 4, any minimizer $(\hat{\theta}_s, \hat{\phi}_{\text{pre}})$ of $L_{\mathcal{D}_s}^\lambda(\theta_s, \phi)$ either satisfies $\hat{\phi}_{\text{pre}} = 0$, or only one $\phi_i$ is non-zero but looks like (up to scaling) $e_j$ for $j \in [k]$. When $\hat{\phi}_{\text{pre}} = 0$, there is

$$\mathbb{E}_{x,y\sim\mathcal{D}_t}\left[\ell(f_{\hat{\theta}_t,\hat{\phi}_{\text{pre}}}(x),y)\right] = \frac{2}{3} > \frac{10}{27}. \tag{51}$$

When only one $\phi_i$ is non-zero but looks like $e_j$ for $j \in [k]$, since all the first $k$ dimensions are equivalent for the source task, with probability $1 - \frac{1}{k}$, this dimension is $j \neq 1$. The target funciton fine-tuned on this $\hat{\phi}_{\text{pre}}$ looks like $f_{\hat{\theta}_t,\hat{\phi}_{\text{pre}}}(x) = \gamma x_j^2$ for some $\gamma \in \mathbb{R}$, so there is

$$\mathbb{E}_{x,y\sim\mathcal{D}_t}\left[\ell(f_{\hat{\theta}_t,\hat{\phi}_{\text{pre}}}(x),y)\right] = \mathbb{E}_{x,y\sim\mathcal{D}_t}\left[(\gamma x_t^2 - x_1^2)^2\right] \tag{52}$$

$$= \gamma^2 \mathbb{E}_{x,y\sim\mathcal{D}_t}\left[x_t^4\right] - 2\gamma\mathbb{E}_{x,y\sim\mathcal{D}_t}\left[x_t^2 x_1^2\right] + \mathbb{E}_{x,y\sim\mathcal{D}_t}\left[x_1^4\right] \tag{53}$$

$$= \frac{2}{3}\gamma^2 - \frac{8}{9}\gamma + \frac{2}{3} \geq \frac{10}{27}. \tag{54}$$

Combining these two possibilities finishes the proof for fine-tuning. Finnaly, setting $\epsilon = min\{\epsilon_0, \frac{10}{27}\}$ finishes the proof of Theorem 1. $\qquad\square$

### B.2 PROOF OF THEOREM 2

**Lemma 4.** *Define the source loss as*

$$L_{\mathcal{D}_s}^{\lambda}(\theta_s, \phi) = L_{\mathcal{D}_s}(\theta_s, \phi) + \lambda \left( \|\theta_s\|^2 + \|\phi\|_F^2 \right).$$

*Then, for any $\lambda > 0$, any minimizer of $L_{\mathcal{D}_s}^{\lambda}(\theta_s, \phi)$ is one of the following cases:*

*(i) $\theta_s = 0$ and $\phi = 0$.*

*(ii) for one $i \in [m]$, $\theta_{si} > 0$, $\phi_i = \pm(\sqrt{2}\theta_{si}) \cdot e_j$ for some $j \leq k$; for all other $i \in [m]$, $|\theta_{si}| = \|\phi_i\| = 0$.*

*Furthermore, when $0 < \lambda < 0.1$, all the minimizers look like (ii).*

*Proof of lemma 4.* Define $d \times d$ matrix

$$A = \sum_{i=1}^{m} \theta_{si} \phi_i \phi_i^{\top}. \tag{55}$$

Define $x_{[1:k]}$ and $x_{[k+1:d]}$ be the first $k$ and last $d - k$ dimensions of $x$. $A_{k,k}$, $A_{k,\bar{k}}$ and $A_{\bar{k},\bar{k}}$ be $k \times k$, $k \times (d - k)$ and $(d - k) \times (d - k)$ matrices that correspond to the upper left, upper right and lower right part of $A$. For a random vector $x$ where the first $k$ dimensions are uniformly independently from $\{\pm 1\}$, the last $d - k$ dimensions are uniformly indepedntly from $\{0, \pm 1\}$, define random variables $A_1 = x_{[1:k]}^{\top} A_{k,k} x_{[1:k]}$, $A_2 = x_{[k+1:d]}^{\top} A_{\bar{k},\bar{k}} x_{[k+1:d]}$. (Note that $x$ is defined on a different distribution than $\mathcal{D}_s$.)

The loss part of $L_{\mathcal{D}_s}^{\lambda}(\theta_s, \phi)$ can be lower bounded by:

$$\mathbb{E}_{x,y \sim \mathcal{D}_s} \left[ \ell(f_{\theta_s, \phi}(x), y) \right] \tag{56}$$

$$= \mathbb{E}_{x,y \sim \mathcal{D}_s} \left[ \left( x_{[1:k]}^{\top} A_{k,k} x_{[1:k]} + 2x_{[1:k]}^{\top} A_{k,\bar{k}} x_{[k+1:d]} + x_{[k+1:d]}^{\top} A_{\bar{k},\bar{k}} x_{[k+1:d]} - y \right)^2 \right] \tag{57}$$

$$\geq \mathbb{E}_{x,y \sim \mathcal{D}_s} \left[ \left( x_{[1:k]}^{\top} A_{k,k} x_{[1:k]} + x_{[k+1:d]}^{\top} A_{\bar{k},\bar{k}} x_{[k+1:d]} - y \right)^2 \right] \tag{58}$$

$$= \frac{2}{3} \mathbb{E} \left[ (A_1 - 1)^2 \right] + \frac{4}{3} \mathbb{E} \left[ (A_1 - 1)A_2 \right] + \mathbb{E} \left[ A_2^2 \right] \tag{59}$$

$$= \frac{2}{3} \mathbb{E} \left[ (A_1 + A_2 - 1)^2 \right] + \frac{1}{3} \mathbb{E} \left[ A_2^2 \right]. \tag{60}$$

The inequality is because

$$\mathbb{E}_{x,y \sim \mathcal{D}_s} \left[ \left( x_{[1:k]}^{\top} A_{k,k} x_{[1:k]} \right) \left( x_{[1:k]}^{\top} A_{k,\bar{k}} x_{[k+1:d]} \right) \right] = 0, \tag{61}$$

$$\mathbb{E}_{x,y \sim \mathcal{D}_s} \left[ \left( x_{[k+1:d]}^{\top} A_{\bar{k},\bar{k}} x_{[k+1:d]} \right) \left( x_{[1:k]}^{\top} A_{k,\bar{k}} x_{[k+1:d]} \right) \right] = 0, \tag{62}$$

$$\mathbb{E}_{x,y \sim \mathcal{D}_s} \left[ \left( x_{[1:k]}^{\top} A_{k,\bar{k}} x_{[k+1:d]} \right) y \right] = 0. \tag{63}$$

The inequality is equality if and only if $A_{k,\bar{k}} = 0$.

The regularizer part of $L_{\mathcal{D}_s}^{\lambda}(\theta_s, \phi)$ can be lower bounded by:

$$\sum_{i=1}^{m} \left( (\theta_{si})^2 + \|\phi_i\|^2 \right) \geq \frac{3}{2^{2/3}} \sum_{i=1}^{m} \left( |\theta_{si}| \cdot \|\phi_i\|^2 \right)^{2/3} \tag{64}$$

$$\geq \frac{3}{2^{2/3}} \left( \sum_{i=1}^{m} |\theta_{si}| \cdot \|\phi_i\|^2 \right)^{2/3} \tag{65}$$

$$\geq \frac{3}{2^{2/3}} \left( \sum_{i=1}^{m} |A_{[i,i]}| \right)^{2/3}, \tag{66}$$

where the first inequality is AM-GM inequality, the second inequality is by concavity of $(\cdot)^{2/3}$. The third inequality is because for diagonal matrix $D$ that has 1 at $(i,i)$ if $A_{[i,i]} \geq 0$, $-1$ at $(i,i)$ if $A_{[i,i]} < 0$, we have

$$\sum_{i=1}^{m} |A_{[i,i]}| = tr(AD) = \sum_{i=1}^{m} \theta_{si} \phi_i^\top D \phi_i \leq \sum_{i=1}^{m} |\theta_{si}| \cdot \|\phi_i\|^2. \tag{67}$$

All the inequalities are equality if and only if $(\theta_{si})^2 = \frac{1}{2} \|\phi_i\|^2 > 0$ for at most one $i \in [m]$, and for all other $i \in [m]$ there is $|\theta_{si}| = \|\phi_i\| = 0$.

Combining the two parts gives a lower bound for $L^\lambda_{\mathcal{D}_s}(\theta_s, \phi)$:

$$L^\lambda_{\mathcal{D}_s}(\theta_s, \phi) \geq \frac{3\lambda}{2^{2/3}} \left(\sum_{i=1}^{m} |A_{[i,i]}|\right)^{2/3} + \frac{2}{3}\mathbb{E}\left[(A_1 + A_2 - 1)^2\right] + \frac{1}{3}\mathbb{E}\left[A_2^2\right] \tag{68}$$

$$\geq \frac{3\lambda}{2^{2/3}} \left(|tr(A_{k,k} + \frac{2}{3}A_{\bar{k},\bar{k}})|\right)^{2/3} + \frac{2}{3}\left(\mathbb{E}[A_1 + A_2] - 1\right)^2, \tag{69}$$

where both inequalitites are equality if and only if $A_{\bar{k},\bar{k}} = 0$ (therefore $A_2 = 0$) and $Var[A_1] = 0$ (therefore $A_{k,k}$ is diagonal by Lemma 5).

Notice that $\mathbb{E}[A_1 + A_2] = tr(A_{k,k} + \frac{2}{3}A_{\bar{k},\bar{k}})$, the above lower bound is further minimized when $\mathbb{E}[A_1] = \mu^*$ where $\mu^*$ is the minimizer of function $L(\mu) = \frac{3\lambda}{2^{2/3}}(|\mu|)^{2/3} + \frac{2}{3}(\mu - 1)^2$.

To see when this lower bound is achieved, we combine all the conditions for the inequalities to be equality. When $\mu^* = 0$, this lower bound is achieved if and only if $\theta_s = 0$ and $\phi = 0$. When $\mu^* > 0$, this lower bound is only achieved when the solution look like this: for one $i \in [m]$, $\theta_{si} = (\frac{\mu^*}{2})^{1/3}$, $\phi_i = \pm(\sqrt{2}\theta_{si}) \cdot e_j$ for some $j \leq k$; for all other $i \in [m]$, $|\theta_{si}| = \|\phi_i\| = 0$.

Obviously, there is either $\mu^* = 0$ or $\mu^* > 0$. Also, when $\lambda < 0.1$, the minimizer $\mu^*$ of $L(\mu)$ is strictly larger than 0 (since $L(1) < L(0)$). So this completes the proof. $\square$

**Lemma 5.** *Let $M \in \mathbb{R}^{k \times k}$ be a symmetric matrix, $x \in \mathbb{R}^k$ is a random vector where each dimension is indepedently uniformly from $\{\pm 1\}$. Then, $Var[x^\top M x] = 0$ if and only if $M$ is a diagonal matrix.*

*Proof of Lemma 5.* In one direction, when $M$ is diagonal matrix, obviously $Var[x^\top M x] = 0$. In the other direction, when $Var[x^\top M x] = 0$, there has to be $x^\top M x$ be the same for all $x \in \{\pm 1\}^k$. For any $i \neq j$, let $x^{(1)} = 1 - 2e_i - 2e_j$, $x^{(2)} = 1$, $x^{(3)} = 1 - 2e_i$, $x^{(4)} = 1 - 2e_j$. Then the $(i,j)$ element of $M$ is $\frac{1}{8}\left(x^{(1)^\top} M x^{(1)} + x^{(2)^\top} M x^{(2)} - x^{(3)^\top} M x^{(3)} - x^{(4)^\top} M x^{(4)}\right)$, which is 0. So $M$ has to be a diagonal matrix. $\square$

*Proof of Theorem 2.* Define the source loss as in Lemma 4, then we have

$$L^\lambda_{\text{meta}}(\theta_s, \phi) = L^\lambda_{\mathcal{D}_s}(\theta_s, \phi) + \mathbb{E}\left[L_{\widehat{\mathcal{D}}_t^{\text{val}}}(\widehat{\theta}_t(\phi), \phi)\right]. \tag{70}$$

By Lemma 4, the source loss $L^\lambda_{\mathcal{D}_s}(\theta_s, \phi)$ is minimized by a set of solutions that look like this: for one $i \in [m]$, $\theta_{si} > 0$, $\phi_i = \pm(\sqrt{2}\theta_{si}) \cdot e_j$ for some $j \leq k$; for all other $i \in [m]$, $|\theta_{si}| = \|\phi_i\| = 0$.

When $j = 1$, the only feature in $\phi$ is $e_1$. When $n_t \geq 18 \log \frac{2}{\xi}$, according to Chernoff bound, with probability at least $1 - \frac{\xi}{2}$ there is strictly less than half of the data satisfy $x_1 = 0$. Therefore, any $\widehat{\mathcal{D}}_t^{\text{tr}}$ contains data with $x_1 \neq 0$, and the only target head that fits $\widehat{\mathcal{D}}_t^{\text{tr}}$ has to recover the ground truth. Hence there is $\mathbb{E}\left[L_{\widehat{\mathcal{D}}_t^{\text{val}}}(\widehat{\theta}_t(\phi), \phi)\right] = 0$.

When $j \neq 1$, the only feature is $e_j$. This feature can be used to fit the target data if and only if either $x_{i[j]}^2 = x_{i[1]}^2$ for all target data $x_i$, or $x_{i[1]} = 0$ for all $x_i$. Since there are at most $k - 1$ possible $j$, by union bound we know the probability of any of these happens for any $j \neq 1$ is at most $k(\frac{2}{3})^{n_t}$.

Hence, when $n_t \geq 3 \log \frac{2k}{\xi}$, the probability of any $e_j$ fits the target data is smaller than $\frac{\xi}{2}$. Therefore, with probabiltiy $1 - \frac{\xi}{2}$, $\mathbb{E}\left[L_{\widehat{\mathcal{D}}_t^{\mathrm{val}}}(\widehat{\theta}_t(\phi), \phi)\right] > 0$ for any $j \neq 1$.

So with probability at least $1 - \xi$, the only minimizer of $L_{\mathrm{meta}}^\lambda(\theta_s, \phi)$ is the subset of minimizers of $L_{\mathcal{D}_s}^\lambda(\theta_s, \phi)$ with feature $e_1$, and with this $\phi$ and any random $\widehat{\mathcal{D}}_t^{\mathrm{tr}}$, the only $\theta_t(\widehat{\mathcal{D}}_t^{\mathrm{tr}}, \phi)$ that fits the target recovers the ground truth, i.e., $\mathbb{E}_{x,y \sim \mathcal{D}_t}\left[\ell_{\theta_t(\widehat{\mathcal{D}}_t^{\mathrm{tr}}, \phi), \phi}(x, y)\right] = 0$.

$\square$

