# OpenReview forum: "Meta-learning Transferable Representations with a Single Target Domain"
_ICLR.cc/2021/Conference — Reject_

### Official Review · AnonReviewer3 · 2020-10-25
**Interesting meta-learning ideas for transfer-learning**

**Rating:** 6
**Confidence:** 3

**Review:**

### Summary
The paper investigates failure cases for transfer learning (fine-tuning and joint training), specifically in the context where training on the source data may highlight features that are irrelevant for the target data. This is done through semi-synthetic data. Based on the insights, the authors present an approach called Meta Representation Learning (MeRLin) inspired by Meta Learning and Learning-to-learn approaches. This approach is evaluated on several real-world transfer-learning tasks from vision and NLP. The authors also derive theoretical results on constructed data distributions for which superiority of Merlin can be shown analytically.

### Quality, clarity, originality and significance

The work is well-motivated, and the paper is well-written and clear.

The "failure modes" and the behavior on the semi-synthetic data are not entirely surprising in my opinion (and I would have expected more analysis after reading the introduction), but they highlight the targeted problem well.

The suggested algorithm is well-motivated and linked to existing work. However in many practical settings, it may be infeasible to train including the whole source data for reasons of compute and data availability, which is why many transfer-learning approaches do not assume the availability of the source data during transfer. This scenario is encountered in the case of BERT, where the authors then "only meta-learn the representation". In that setting, Merlin seems to reduce to reduce to a changed objective inspired by meta-learning and learning-to-learn. This is not a problem in itself, but it makes it less clear, what exactly the authors think that the main contribution of the paper is.

The experimental results look good at first glance, but overall I found it hard to evaluate how convincing they are because of several potential problems:
* The experimental results are not compared to any results from the existing literature, all baselines and comparisons are entirely from this paper. This makes it hard to know how much careful tuning went into the proposed algorithm versus the baselines. I think at least *some* comparison to results from the literature should be possible or it should be carefully explained why such an overfitting to the proposed algorithm clearly did not happen.
* There are a lot of seemingly random choices in the selection of the datasets, e.g. why USPS and not MNIST, why CUB/Caltech/Cars and not e.g. CIFAR, Pets, Flowers, or Birds? I'm not saying any of these choices are inherently better, but the authors should explain *why* these datasets are chosen, to avoid the impression that the datasets may have been chosen because the proposed method works particularly well on these dataset combinations. One way to make the results stronger would therefore be to use combinations of datasets that other, previous work has already used, which would also enable a direct comparison (see above).
* Similarly, it is unclear why only a subset of GLUE was used and not the full GLUE benchmark. Again, it would be better in my opinion to use all of GLUE to avoid the impression that the subset was hand-picked. Also, I tried to find results on GLUE for direct comparison, because they should exist, but was not successful immediately, because the presented results are on the dev-set (Table 2) while usually results for comparison come from the official test set as far as I understand? E.g. the BERT paper seems to have tuned on dev and then reported numbers on test, but I did not find dev-set numbers in a quick search. For something fairly standard as BERT-base and GLUE I think it would be good to be able to find numbers in the literature that are *directly comparable*.
* Again in a similar fashion, Sec.6.3. evaluates only two target data sets, why only these? And A.5 looks only at Food->Cub, why only this pair?

### Pros and cons
* Pros: interesting approach, well-motivated, well-written, interesting theoretical analysis
* Cons: experimental results are not entirely convincing, the main message is not 100% clear to me

### Minor details and comments
* The paper is fairly "squeezed": E.g. lack of vertical space after captions of Figures 1 and 2; also the main suggested algorithm is not explained in detail in the main text, but deferred to the Appendix. It might be more readable to try to shorten the content.
* "Pre-training only" is not a complete transfer-learning algorithm because of potential class number mismatch, yet it is used for comparison (e.g. Fig. 2b). This only works in this very specially constructed scenario, it seems. Maybe this should be mentioned?
* "Target-only" is discussed in the context of Fig. 2 but it is not contained in the Figure, why? Also, "fine-tuning" results are not shown in Fig 2b.
* Towards the end of Section 3, when discussing the synthetic data, "overfitting" is mentioned twice as a problem. There exist several approaches that are often used to try to overcome overfitting, and the authors even use L2-sp as a baseline in Section 6. It remains a bit unclear how much a careful combination of fine-tuning and such methods could help here. I don't think this is a major shortcoming, though, because the artificial setup is constructed exactly in such a way that the pre-trained classifier will focus on the non-transferable features.
* Table 1: For ImageNet->C256 L2-sp seems to be not clearly worse than MeRLin given the standard deviations of three runs. Maybe both should be bolded?
* A.7 "the model is not sensitive to varying ρ and λ." - it was not clear to me how much this statement can be derived from the figure.

* Minor comments / typos:
  * Caption of Fig.1 - I did not see how this is an illustration of the "features learned"
  * End of Sec.3: "on a much simplified settings"
  * page 5: "Merlin to by changing" - "to" too much
  * page 5: "with AB as the source ." - space before "."
  * page 6: "with their own best regularization strength" - I did not understand this phrase here.
  * before Eq.(4) - what is $\Omega$ here?
  * In Theorems 1 and 2, what exactly is meant by "universal constant"?
  * after Eq.(6): "because its simultaneously fits" -> it
  * Food-101->CUB: here, the model is not mentioned. I guess it was also ResNet-18?
  * page 7 "join training" -> joint
  * page 8 "the our method"
  * References: some letters in the reference titles are lowercased when they shouldn't be, e.g. "cnns", "Mask r-cnn", "reuse? towards", "t-sne", "mnist", "Xlnet". Also, several references do not contain a "publishing venue", e.g. Kolesnikov 2019, Lin 2002, Liu 2019a, Neyshabur 2020 - in the age of search engines this is not a real problem, but it might be nice to give at least an arXiv number if available or some of these have probably appeared in conferences or journals.
  * page 12: "images is of resolution"
  * A.2 "lambda is found with cross validation" - how exactly?
  * A.4 for completeness maybe include ImageNet and USPS?
  * Figure 4 caption: "textbfSensitivity"

---

> ### Author Response · Authors · 2020-11-18
> **Response**
>
> We thank the reviewer for the comments. We addressed the concerns and will further incorporate the feedback in our revision.
>
> 1.  Comparison to results from existing literature and choices of datasets
>
> The choice of using CUB, Caltech and Cars as the target datasets with ResNet-18 was purely because they are popular choices in related work (L2-sp, [1] and [2]). We chose to experiment in addition on Food->CUB to validate the setting where the discrepancy between two domains is large. Here we further provide results on Stanford Dogs, Aircraft, and MIT-indoors (also popular with results in [1] and [2]) with ResNet-50 in the revision and compared with results provided in [1] and [2] (See the table below) to address the concern. We add these results to Table 3 of the revision. We also provide the SVHN or Fashion-MNIST -> MNIST results.
>
> Top-1 Accuracy on ResNet-50 with imageNet as the source dataset.*
>
> | Method | MIT-indoors | Stanford Dogs  |   Aircraft |
> |---|:---:|:---:|:---:|
> | Fine-tuning  |  83.21 ± 0.21  | 84.77 ± 0.32   |  81.13 ± 0.21 |
> | L2-sp      | 83.98 ± 0.29 | 86.42 ± 0.20  |  80.98 ± 0.29  |
> | BSS [1]   |  83.73 ± 0.18 |  86.69 ± 0.19  | 81.48 ± 0.18  |
> | Delta [2]    |    83.66 ± 0.20   |   85.01 ± 0.22   |  80.44 ± 0.20 |
> |  MeRLin   |   84.50 ± 0.26  |    87.05 ± 0.33   |  82.57 ± 0.27 |
>
> *L2-sp, Delta, and BSS results are from [1] and [2].
>
>
> Top-1 Accuracy with LeNet on MNIST (600).
>
> |Source |  SVHN  |  Fashion-MNIST|
> |---|:---:|:---:|
> |Target Only    |     92.74 ± 0.63   |   92.74 ± 0.63 |
>  |  Fine-tuning       |     93.22 ± 0.28 |  92.82 ± 0.25 |
>  |  DANN [4]       |        92.92 ± 0.34  |   92.61 ± 0.29 |
>  |  MeRLin      |           95.20 ± 0.15   |   94.87 ± 0.14 |
>
> [1] Xinyang Chen, Sinan Wang, Bo Fu, Mingsheng Long, Jianmin Wang, Catastrophic Forgetting Meets Negative Transfer: Batch Spectral Shrinkage for Safe Transfer Learning, NeurIPS 2019
>
> [2] Xingjian Li，Haoyi Xiong，Hanchao Wang，Yuxuan Rao，Jun Huan, DELTA: DEEP LEARNING TRANSFER USING FEATURE MAP WITH ATTENTION FOR CONVOLUTIONAL NETWORKS, ICLR 2019
>
> [3] Barret Zoph, Golnaz Ghiasi, Tsung-Yi Lin, Yin Cui, Hanxiao Liu, Ekin D. Cubuk, Quoc V. Le, Rethinking Pre-training and Self-training, arxiv 2006.06882
>
> [4] Domain-adversarial training of neural networks, JMLR 2017

---

> > ### Comment · AnonReviewer3 · 2020-11-23
> > **Thank you for the response**
> >
> > Thank you for your response and updates.
> >
> > I agree with AnonReviewer5 on the still remaining point of distinction to "traditional transfer-learning" (use of source training data during transfer), which was raised by all three reviewers in their comments and which you suggest to clarify further "in the next revision".
> >
> > I  would also like to note that some of the points I mentioned seem to not have been addressed in your response/update, e.g.:
> > * Can there be comparisons to test-set/dev-set numbers for GLUE sub-tasks?
> > * Sec.6.3. evaluates only two target data sets, why only these?
> >
> > Thank you also for including more comparison results in your revision. I tried to find the numbers in your table in the references you point to, but was only partially successful. E.g. in the [CSS paper](http://papers.neurips.cc/paper/8466-catastrophic-forgetting-meets-negative-transfer-batch-spectral-shrinkage-for-safe-transfer-learning.pdf) I was able to find the 81.48 on Aircraft, but not the other two numbers, maybe I overlooked something? Similarly in the [DELTA paper](https://arxiv.org/pdf/1901.09229.pdf) I saw only numbers for ResNet-101, not for ResNet-50. Maybe I'm looking at the wrong references?

---

> > > ### Author Response · Authors · 2020-11-24
> > > **Thanks for the response**
> > >
> > > Thanks for reading our response and asking the new questions.
> > >
> > > 1.	We agree that the setting of this paper is different from pre-training and fine-tuning. As we have shown in the paper both theoretically and empirically, pre-training can learn source-specific features since it has no access to the target dataset. Thus, without the source data it’s difficult to do better than fine-tuning. Besides, the setting of our paper is also practical, since we usually have access to source data such as ImageNet. Also note that self-training and joint training also require source data. We will elaborate more on this point and clarify the difference between standard fine-tuning and our setting in the next revision.
> > >
> > > 2.	We provide results on GLUE test-set of the three sub-tasks in the following table.
> > >
> > > Results on GLUE test-set
> > >
> > > | | MRPC | RTE | QNLI |
> > > | --- | :---: | :---: | :---: |
> > > | Fine-tuning | 88.9 | 66.4 | 90.1 |
> > > | MeRLin | 89.5 | 68.9 | 90.7 |
> > >
> > > MeRLin outperforms standard fine-tuning, but we only have one run of each result due to the limitation on number of submissions of the GLUE server.
> > >
> > > 3.	In Sec.6.3, we run the experiments on the two datasets only because we have limited computational resources. We will add results on more datasets in the next revision.
> > >
> > > 4.	We accidentally compared with the earlier conference version of BSS (which was shared by the authors via private communication). The Delta results are also provided from the BSS paper. In the camera-ready version, they have removed the MIT-indoors numbers. The earlier conference version with MIT-indoors results can be found in https://drive.google.com/file/d/155H0gxZrnb_ZdgQjfKYx5JctBdffCEPv/view?usp=sharing (an anonymous google drive link).

---

> > > > ### Comment · AnonReviewer3 · 2020-11-24
> > > > **Thank you for the clarifications**
> > > >
> > > > Thank you for the clarifications!

---

> > > > ### Author Response · Authors · 2020-11-25
> > > > **Further comparison of BSS and MeRLin**
> > > >
> > > > We also add results of BSS on ResNet-18 to compare with our method in the following table.
> > > >
> > > > |Method|CUB-200|Caltech-256|Stanford Cars|
> > > > | --- | :---: | :---: | :---: |
> > > > |Fine-tuning|72.52 ± 0.51|81.12 ± 0.27 |81.59 ± 0.49|
> > > > |BSS|73.43 ± 0.21|82.21 ± 0.18|81.84 ± 0.25|
> > > > |MeRLin|75.42 ± 0.47 | 82.45 ± 0.26|83.68 ± 0.57|
> > > >
> > > > MeRLin achieves more significant improvement over fine-tuning compared with BSS.

---

### Official Review · AnonReviewer4 · 2020-10-28
**Insightful study and reasonable method; while experiment and related works can be further polished**

**Rating:** 6
**Confidence:** 4

**Review:**

This paper first investigates when and why fine-tuning and joint training are not the best methods for transfer learning. The authors generate a toy dataset, in which the source-specific and transferable features are thus clearly distinguishable, for the transfer learning tasks. As both fine-tuning and joint training are not optimal solutions for the experimental dataset, they further design MeRLin, which is based on the meta-learning mechanism, to learn a generalizable model weight.

*Strengths*:
1. In order to investigate how a model (including a feature extractor and a classifier) interacts with the source-specific feature and transferable feature within a data, the authors generate a toy dataset and examine the effectiveness of fine-tuning and joint-training methods. This provides clear and insightful experimental observations for the further understanding of transfer feature learning.

2. The observation that the fine-tuning and the joint training tends to learn the "convenient" feature (i.e., both methods tend to learn easy source-specific feature rather than a transferable feature) provides a reasonable argument why both models cannot work well.

3. The designed meta-learning-based method is convincing empirically, as the objective function implies the model has to learn a generalizable feature extractor for loss minimization.

*Weakness/comments*
1. Although the proposed demonstrates the above observations, the connection to existing domain adaptation/domain generation works seems lacking. Some of these works also aim to learn the feature, which can be generalized to the target domain. And the source-specific feature can be also regarded as the domain shift between source and target domains (so that it cannot be generalized by using fine-tuning). I understand the problem settings of domain adaptation/domain generation are a bit different from the setting used in the paper, while such discussion can further improve the completeness of this paper.

2. Another weakness is the baseline methods, as these baseline methods are too simple. I am wondering about the comparison to existing works on the relevant areas (e.g., other general feature learning or domain-relevant works).

3. One reason why people use pretraining method is neglected: one does not need to access the source dataset when the target task is trained. In some practical scenarios, one cannot access the large scale of source dataset (i.e., imagenet), and this limits the use of the proposed model, which requires access to source data. But this point is just a comment, not a weakness as the authors have clarified the problem scope at the beginning of the paper.

Overall, the paper presents an insightful study and a reasonable new method. I am inclined to the score "Marginally above acceptance threshold".

---

> ### Author Response · Authors · 2020-11-18
> **Response**
>
> Thanks for the comments. The reviewer noted that our experiments “provide clear and insightful experimental observations for the further understanding of transfer feature learning”, and the proposed method is “convincing empirically”. We address the comments and will incorporate them in the revision.
>
> 1.“Connection to domain adaptation (DA) / domain generalization (DG) literature.”
>
> We thank the reviewer for the comment. We note that DA and DG methods generally require the same label set for the source and target domains. In contrast, we worked in the setting where the source and target domains have different label sets (or even different tasks). For example, in the ImageNet -> CUB200 the source dataset has 1000 classes but the target dataset has 200 classes that are different from the source.  We will add further discussions to the revision to clarify the difference between domain adaptation and the transfer learning setting we studied. For completeness, we also provide results of DANN [4] (a popular DA method) on SVHN or Fashion-MNIST -> MNIST where the label sets are the same. (See Table below.)
> Top-1 Accuracy with LeNet on MNIST (600).
>
> |Source   |     SVHN   | Fashion-MNIST|
> |---|:---:|:---:|
> |Target Only |    92.74 ± 0.63 |  92.74 ± 0.63|
> |Fine-tuning     |    93.22 ± 0.28   |92.82 ± 0.25|
> |DANN [4]        |    92.92 ± 0.34  |92.61 ± 0.29|
> |MeRLin            |  95.20 ± 0.15  | 94.87 ± 0.14|
>
> 2. “comparison to  other general feature learning or domain-relevant works”
>
> We applied DANN [4] (a popular DA method) to SVHN or Fashion-MNIST -> MNIST. We added the domain confusion loss of DANN to the objective of joint training. DANN does not improve the performance when we have access to target labels. (Although we note that DANN was originally designed for unsupervised domain adaptation where no unlabeled data are available. Here we add the target label loss with DANN loss together for comparison in our setting.)
>
> [4] Domain-adversarial training of neural networks, JMLR 2017

---

### Official Review · AnonReviewer5 · 2020-11-10
**A strong piece of theory and and principled analysis which gives way to an algorithm with questionable benefits**

**Rating:** 5
**Confidence:** 3

**Review:**

## Summary of the Work
The work analyzes the failure modes of fine-tuning and joint training for multi-task learning in a computer vision setting, then proposes a meta-learning algorithm dubbed "MeRLin" to address these failure modes using bi-level optimization. The outer optimization minimizes the loss of inner-trained network when it is used to learn a new classification head for a validation set from the target domain.

The authors first motivate their work by discussing the success and failures of joint training and fine-tuning, then they use principled experiments to verify hypotheses about the failure modes of these two popular transfer methods. Thereafter, they introduce their proposed method (MeRLin), provide a theoretical analysis which compares its behavior to fine-tuning and joint training in a limited problem class, and then present empirical results showing the effectiveness of MeRLin on several popular CV and NLP benchmarks.

## Pros and Cons

### Pros
* Principled approach to analyzing the failure modes of fine-tuning and joint training


### Cons
* Requiring data from the target domain during training negates many of the sought-after benefits of transfer learning (e.g. re-using pre-trained models quickly without lengthy pre-training, adapting to unforseen new tasks, etc.)

## Evaluation
### Quality
3/5
The overall quality of execution of this work is high, but not flawless.   The work's laudable careful theoretical and synthetic-empirical evaluations give way to less-careful statistical comparisons of the empirical-nonsynthetic results. I believe works making specific performance claims based on empirical experiments without exception need to provide specific evidence in the form of statistical tests of significance, especially in cases like these when the performance gaps between the proposed method and baselines are so small.

### Clarity
5/5
The text and diagrams themselves are clean, lucid, well-written, and well-presented. I don't personally find the analysis in Sec. 5 to be particularly helpful to understanding the method or the failure modes of fine-tuning and joint training, and would be happy to see it moved to the appendix in favor of more careful empirical evaluation, but perhaps some readers find it easier to understand concepts with theory.

### Originality
4/5
This work has clear relationships to many prior works on parameter-space meta-learning and automatic hyperparameter tuning, and its own spin on the problem, which the authors acknowledge and provide context for. I have erred on the side of a higher rating in this category, thought don't have encyclopedic experience with all of the background literature. While the proposal in the work is not eye-wateringly novel considering previous work on regularization and meta-learning, novelty in ML research is, in general, overrated, and I believe this work is sufficiently novel to be a contribution.

### Significance
2/5
The analysis and synthetic experiments in this work are, I believe, a significant-enough contribution to the field irrespective of the proposed algorithm. If the paper itself contained only extended versions of the theoretical and synthetic-empirical experiment portions, I think it would be significant enough.

Unfortunately, I have concerns about the significance of this work--inclusive of the proposed algorithm, which readers will focus on--for 2 reasons:
(1) The proposed transfer solution negates many perceived and desired benefits of transfer and meta-learning, because it requires access to samples from the target distribution at train-time. Specifically, most researchers and practitioners pursue transfer learning specifically because they anticipate *not* having access to target samples at train time, e.g. in the case of adapting CV classifiers to unforseen target samples, or in the case of re-using a very large pre-trained CV or NLP model which--might be impractical for the end-user to train--for some unforseen application by an end user. The work does not address or acknowledge at all this slight compromise of the problem setting, or make efforts to justify its practicality in spite of this.
(2) While the synthetic experiments show great performance benefits to the proposed method in a fabricated setting, the empirical-non-synthetic experiments show very little improvement over the baselines, especially L2-sp (Li, et al. 2018), and it's not clear from the provided analysis that the performance differences presented are even statistically-significantly different from L2-sp. This is especially notable given that fact that L2-sp *does not* require access to target samples at train-time (i.e. it conforms to the more-general transfer learning setting), while MeRLin compromises the setting to achieve apparently-comparable but not substantially-improved results.

### Misc Editorial Comments and Reviewer's Notes

#### Claims
* The reader will learn about "when and why fine-tuning and joint training can be suboptimal or even harmful for transfer learning"
* Proposed method MERLIN empirically outperforms SOTA transfer learning algorithms on vision and NLP benchmarks
* Proves that MERLIN recovers the target ground truth model under some assumptions on NN architecture and source distributions, and that pre-training and join training fail


#### Mechanisms
* Ensuring that a head fit on top of the learned representations with (target) training data also performs well on (target) validation data


#### 1. Introduction
* Does the assertion "joint training easily overfits to the target especially when the target dataset is small" hold if the target dataset is intentionally oversampled so that its examples appear as often as the source dataset? If we don't have the answer to this question, it could be that we are simply observing the effect of the data distribution being skewed.
* "any proposed algorithm [...] should use the source and target simultaneously..." -- I think this statement is overly-strong considering the results in the paper. Certainly this is *one way* to ensure that we "extract shared structures," but it also presupposes we have the target up front, which is opposed to many sought-after benefits of a transfer learning approach. Using the same evidence, we can argue instead that transfer learning methods should make use of as much data as possible, or search for data sources which are more diverse and general than both the source and the target.

#### 5. Theoretical Analysis
* I find this lucid but perhaps unnecessary for the reader's understanding of the concept. It seems fairly intuitive that a learning algorithm with no access, or unbalanced access to to the target distribution will fail to learn features which generalize to that target distribution, and a learning algorithm which is intentionally supervised, using a loss computed on samples from the target distribution, will learn features which generalize to both the source and the target. *The important question then is "do we give up more than we gain by requiring access to the target distribution?"* I recommend moving this portion to the Appendix to make more room for justifying the method and better in-text coverage of empirical results.

#### 6. Experiments
* Many of these results would seem to be quite close, especially when using only 3 trials and simple mean/stddev. How do the results compare with a real confidence interval? e.g. a 95% bootstrapped interval (note the seaborn package will easily calculate these for you)? Are the differences observed statistically significant?

---

> ### Author Response · Authors · 2020-11-18
> **Response (continued)**
>
> 4) “Does the assertion "joint training easily overfits the target especially when the target dataset is small" hold if the target dataset is intentionally oversampled so that its examples appear as often as the source dataset?”
>
> In practice we use stronger data augmentation for the target domain and tune the hyper-parameter \alpha (the coefficient in front of the target loss) to overcome the imbalance between the source and target domain. A larger \alpha corresponds to reweighting up the target data.  We also show in the theoretical analysis even with optimal \alpha, joint training still overfits the target dataset. We suspect resampling the target will also not work because it has a somewhat similar effect to reweighting. This is also congruent with the findings in the literature on imbalanced dataset (e.g., [5]), which shows that re-sampling or reweighting rare data can potentially make the rare data train better, but do not solve the poor generalization performance of the rare data.
>
> [1] Xinyang Chen, Sinan Wang, Bo Fu, Mingsheng Long, Jianmin Wang, Catastrophic Forgetting Meets Negative Transfer: Batch Spectral Shrinkage for Safe Transfer Learning, NeurIPS 2019
>
> [2] Xingjian Li, Haoyi Xiong, Hanchao Wang, Yuxuan Rao, Jun Huan, DELTA: DEEP LEARNING TRANSFER USING FEATURE MAP WITH ATTENTION FOR CONVOLUTIONAL NETWORKS, ICLR 2019
>
> [3] Barret Zoph, Golnaz Ghiasi, Tsung-Yi Lin, Yin Cui, Hanxiao Liu, Ekin D. Cubuk, Quoc V. Le, Rethinking Pre-training and Self-training, arxiv 2006.06882
>
> [4] Domain-adversarial training of neural networks, JMLR 2017
>
> [5] Mengye Ren, Wenyuan Zeng, Bin Yang, Raquel Urtasun, Learning to Reweight Examples for Robust Deep Learning, ICML 2018

---

> > ### Comment · AnonReviewer5 · 2020-11-21
> > **Thank you**
> >
> > Thank you for your thoughtful responses and updates! I will take them into consideration and have no further questions.
> >
> > I will reiterate that, because this setting is significantly different from what most readers will expect from the title, I believe it should be highlighted, amply-discussed, and justified as an important setting in the manuscript to a much larger degree than is currently represented. Otherwise, this work runs a very high risk if misleading readers with a title which makes them expect a traditional transfer-learning setting.

---

> > > ### Author Response · Authors · 2020-11-21
> > > **Thanks for reiterating this point**
> > >
> > > Thanks for reading our response and for reiterating this point. We agree that we should highlight more about the setting and justify its applications to avoid any potential misinterpretation. We will definitely do this in the next revision.

---

> ### Author Response · Authors · 2020-11-18
> **Response**
>
> We thank the reviewer for the detailed and constructive comments. The reviewer noted that our theoretical analysis is “strong” and the semi-synthetic analysis of transfer learning algorithms is “principled”. We fully address the concern and will incorporate them in the revision.
>
> 1) “The proposed transfer solution negates many perceived and desired benefits of transfer and meta-learning, because it requires access to samples from the target distribution at train-time.”
>
> Thanks for the comments and the observation. We agree that we require both the source and target data in the training time whereas fine-tuning only requires a pre-trained model and the target data. We agree that ideally we want the algorithms to access as few data as possible, but we think the setting that we focused on is also interesting because there are also many realistic situations when the people who have the target data (e.g., bioimaging data in a hospital) can also access a publicly available source dataset (e.g., ImageNet.) As Reviewer #4 noted, this is “not a weakness as the authors have clarified the problem scope at the beginning of the paper”.
>
> Moreover and perhaps more importantly, in the paper, we argue with evidence that in order to achieve substantial improvement over fine-tuning, we need to access both the source data and target data together.  As motivated in the introduction and shown theoretically in Sec 5 and empirically in Sec 3, standard pre-trained models can learn source-specific features instead of transferable features. When this happens, the pretrained models provide insufficient knowledge for the target. Similar phenomenon can be found in [3], which compares pre-training with self-training and finds that the access to target data when training on the source domain does help self-training perform better than pre-training + fine-tuning.
>
> 2) “the empirical-non-synthetic experiments show very little improvement over the baselines, especially L2-sp (Li, et al. 2018)”
>
> We would like to clarify that fine-tuning is a strong baseline itself and it’s very challenging to improve upon it. Prior works such as L2-sp, BSS [1], and Delta [2] generally have small improvement over finetuning and sometimes even hurts the performance on some datasets. For example, in eight of the nine settings in Table 1 and Table 2 of the paper, our improvement over L2-sp is (a few times) bigger than the difference between L2-sp and fine-tuning, and in several cases, L2-sp is worse than fine-tuning.
>
> The average improvement of our algorithm over fine-tuning across all datasets is 2.32%, whereas the average improvements of the previous best method L2-sp’s improvement over finetuning is 0.23%. Thus we consider it more significant than that of L2-sp.
>
> Moreover, we also conducted three new experiments on MIT-indoors, Stanford Dogs, and Aircraft with ResNet 50, and compared it with two additional baselines BSS [1] and Delta [2]. The table below corroborates our claim above that prior works have limited improvements over finetuning, whereas our method has significant and consistent gains. We add these results to Table 3 of the revision.
>
> Top-1 Accuracy on ResNet-50 with imageNet as the source dataset. *
>
> | Method | MIT-indoors | Stanford Dogs  |   Aircraft |
> |---|---|---|---|
> | Fine-tuning  |  83.21 ± 0.21  | 84.77 ± 0.32   |  81.13 ± 0.21 |
> | L2-sp      | 83.98 ± 0.29 | 86.42 ± 0.20  |  80.98 ± 0.29  |
> | BSS [1]   |  83.73 ± 0.18 |  86.69 ± 0.19  | 81.48 ± 0.18  |
> | Delta [2]      |      83.66 ± 0.20   |   85.01 ± 0.22   |  80.44 ± 0.20 |
> |  MeRLin     |        84.50 ± 0.26  |    87.05 ± 0.33   |  82.57 ± 0.27 |
>
> *L2-sp, Delta, and BSS results are from [1] and [2].
>
> 3) “it's not clear from the provided analysis that the performance differences presented are even statistically-significantly different from L2-sp.”
>
> We admit we only did 3 seeds for these experiments due to limited computing resources. We provide additional empirical evidence to show that the performance differences are statistically significant. First, we run additional t-tests on SVHN -> USPS, Fashion -> USPS, ImageNet -> CUB-200 and ImageNet -> Stanford Cars with 10 seeds to get a more accurate comparison between L2-sp and our method. The p-values shown below indicate that our improvement over L2-sp is statistically significant. We also add the 95% bootstrapped confidence interval in Section A.8 (Figure 5) of the revision. Second, in all but one of the nine settings shown in Table 1, the difference between the mean performance of L2-sp and our method is bigger than 3 times the standard deviations of them, and in a few of the cases, more than 5 times larger than standard deviation.
>
> t-test results of L2-sp and MeRLin
>
> | Task   |    SVHN -> USPS  |  Fashion -> USPS   |  ImageNet -> CUB-200  |   ImageNet -> Stanford Cars  |
> |---|---|---|---|---|
> |  p-value     |   0.002953         |        1.136e-7          |        1.470e-5                |              2.399e-4   |

---

### Decision · Program_Chairs · 2021-01-07
**Final Decision**

**Decision:**

Reject

**Comment:**

The paper compares transfer learning with fine-tuning and joint training and then proposes a new approach (Merlin). Reviewers have pointed to the fact that Merlin works in a setting that is different from normal transfer learning settings (it assumes some target domain data is available during training). The authors acknowledge this and think it can still be a reasonable setting, but of course it makes comparisons more difficult. Overall, while there are interesting analysis and results, the paper remains borderline and more work should be done to make it a good contribution, including significantly improving the presentation to make clear the distinction in settings. I therefore recommend to reject the paper.